
# Assessment of coastal inundation triggered by multiple drivers in Ca Mau Peninsula, Vietnam

Hung Nghia Nguyen[1], Quan Quan Le [1,2], Dung Viet Nguyen[1,3], Tan Hong Cao[1], Toan Quang To[1], Hai Do Dac[1], Melissa Wood[4,5], Ivan D. Haigh[4]

[1]Southern Institute of Water Resources Research, 658[th] Vo Van Kiet Avenue, District 5, Ho Chi Minh City, Vietnam

[2]Energy and Environment Institute, University of Hull, Hull, HU6 7RX, UK

[3]GFZ German Research Center for Geosciences, Section Hydrology, Potsdam 14473, Germany

[4]School of Ocean and Earth Science, University of Southampton, Waterfront Campus, European Way, Southampton, SO14 3ZH, UK
[5]Marine Systems Modelling Group, National Oceanography Centre, Joseph Proudman Building, 6 Brownlow Street, Liverpool, L3 5DA, UK

*Correspondence to:* Hung Nghia Nguyen (hungsiwrr@gmail.com)

## Abstract

The Ca Mau Peninsula plays a critical role in the agricultural and aquaculture productivity of the Vietnam Mekong Delta (VMD), central to regional food security and the population's economic and social welfare. Unfortunately, this region has also historically been a hotspot for natural disasters, particularly from flooding, which is initiated by seasonal river flux upstream and heightened sea levels downstream, but also exacerbated by global climate change (e.g., increased rainfall and sea-level rise, tropical storm surges) and human activities (e.g. river bed lowering, land subsidence). The potential risks associated with rising inundation levels is important information for the future sustainability of the region and its ability to adapt to both current and forthcoming changes. The research around the influence of such drivers on future flood risk, in the Ca Mau Peninsula, is incomplete, primarily due to the absence of a quantitative coastal inundation map corresponding to future compounded scenarios. In this study, we therefore evaluate flooding dynamics in the Ca Mau peninsula using a fully calibrated 1D model, to represent a range of anthropogenic and climate change compound scenarios. Our findings indicate that factors such as increased high-flows upstream, alterations in the riverbed of the main Mekong channel, and occurrences of storm surges effecting the mainstream Mekong River, are unlikely to significantly affect inundation dynamics in this region. However, land subsidence, rising sea levels, and their combined effects emerge as the primary drivers behind the escalation of inundation events in the Ca Mau peninsula, both in terms of their extent and intensity, in the foreseeable future. These results serve as vital groundwork for strategic development and investment as well as for emergency decision-making and flood management planning, providing essential insights for shaping development policies and devising investment strategies related to infrastructure systems in an area, which is rapidly developing.



**Keywords:** Ca Mau peninsula, Mekong delta; inundation risk, sea level rise, land subsidence, delta vulnerability; floods


## 1 Introduction

Deltas, home to over 500 million people globally, can be important regions for agriculture and food production due to their fertile soils (Pont et al., 2002; Attaher et al., 2009; Edmonds et al., 2020). These low-lying regions, between river and coast, are highly susceptible to flooding, which can be further exacerbated by impacts of climate change (e.g. sea level rise,

atmospheric changes), and human activities such as water abstraction (linked to land subsidence), river damming, and sand dredging rise (Syvitski et al., 2009; Giosan et al., 2014; Bevacqua et al., 2019; Best, 2019; Edmonds et al., 2020). According to current climate change projections, future flooding is expected to intensify in delta regions worldwide with coastlines facing heightened flood risks due to an increased frequency of extreme precipitation events and storm surges (IPCC, 2023).

Current research around delta flooding predominantly focuses on single drivers of flooding, from rivers or from the sea, with

less emphasis on the combined hazard from multiple flood drivers occurring at the same time. Several studies underscore the importance of compound flooding, demonstrating that the impact of combined flood sources can surpass the sum of their individual effects (e.g., Leonard et al., 2014; Paprotny et al., 2018; Ward et al., 2018; Wahl et al. 2018; Ganguli and Merz, 2019). For instance, Olbert et al., (2017) utilized hydrodynamic modelling to assess compound flood mechanisms in Cork City, Ireland, considering tides, storm surges, river fluxes, and their interactions. The study revealed that while high sea levels

alone may not induce flooding, their elevation can exacerbate flooding when coinciding with river flooding. Similarly, Leonard et al., (2014) showed an increasing inundation change when significant river discharges coincided with extreme sea levels, compared to each source acting independently. Chen and Liu, (2014) employed a three-dimensional hydrodynamic model to study inundation in the Tsengwen River basin, southern Taiwan, under the influence of storm surge, freshwater discharge, and their combination. Their findings indicated that the combination of extreme storm surge and high upstream river discharges

could worsen flood severity compared to individual drivers.

These studies underscore the crucial necessity of assessing the future implications of compound flooding for emergency decision-making, flood management, and planning. It is imperative to accurately estimate and quantify compound flood risk, to facilitate proactive mitigation and adaptation measures. A comprehensive understanding of both current and anticipated combined flood hazards enables the implementation of effective engineering solutions, appropriately budgeted to mitigate

flood-related fatalities, operation and construction of technical infrastructure, environmental degradation, and societal impacts (Leonard et al., 2014; Haigh et al., 2014; Wahl et al., 2018; Ward et al., 2018; Paprotny et al., 2018; Edmonds et al., 2020). Moreover, at the emergency planning level, it is essential to consider the impacts of compound flooding to optimize flood protection infrastructure and levee systems (Leonard et al., 2014, Ward et al., 2018).



For the Vietnam Mekong delta (VMD), current research provides insight into the present challenges that exacerbate flood risk
in the region. Recently, river damming and sand mining has resulted in lowering of the riverbed in the VMD (Kummu et al.,
2010; Lu et al., 2014; Bravard et al., 2013; Hackney et al., 2021). It is a major concern because of its potential to affect the
hydraulic regime. Vasilopoulos et al., (2021) estimated that two principal channels of the Mekong delta (the Mekong and
Bassac Rivers) experienced riverbed lowering of an average of 2.5 m ($\sigma = 3.9$ m) between 1998 and 2018. Riverbed lowering
also directly impacts the extent of tidal intrusion penetrating inland, potentially contributing to greater tidal inundation risk
(Eslami et al., 2019). Within the broader VMD area, the vertical sinking of land is a recent phenomenon now occurring at
rates approximately around 0.03 m per year (Erban et al., 2014). This subsidence is expedited by the extraction of groundwater
linked to agricultural practices and the expanding population within the delta (Erban et al., 2014; Minderhoud et al., 2020).
For the future VMD, the report of IPCC, (2007) considers that the VMD will be one of the zones that will be most affected
by climate change by the end of this century. NASA, (2021) suggest that under scenario SSP5-8.5, the VMD could experience
a mean sea level rise (SLR) of 1 m by the end of the century. This level of sea level rise could contributing to the threat of
greater flood magnitudes for the region (Västilä et al., 2010; Lauri et al., 2012; Hoang et al., 2016; MONRE, 2016; Minderhoud
et al., 2019). According to Minderhoud et al., (2019), approximately 51% of the VMD plain would be submerged if the mean
sea level rose by 0.8 m. Wood et al., (2023) highlighted a significant uptick in extreme storm surges anticipated to this coastline
by year 2050, linked to the heightened occurrence of intense tropical cyclones. Van et al., (2012) used 1D modelling to study
the impacts of climate change on future flood risk and showed the downstream area of VMD inundation will be more
aggravated in terms of higher magnitude and longer flooding durations because of the compounding effects of increasing high
tide and fluvial floods. Triet et al., (2020), used a quasi-two-dimensional hydrodynamic model to estimate the future flood
regime of VMD. They found that SLR and land subsidence would present the highest impact driving flooding in the VMD,
with flooded areas extended by around 20-27% in the future period (2036-2065) compared to the past period 1971-2000.

Despite these significant findings, prior research has yet to direct attention to the Ca Mau Peninsula, an area within the Mekong
Delta that stands as the most susceptible region (see Fig. 2a for exact locations). Furthermore, prior research has used
bathymetry data collected from a range of sources (introducing inconsistencies and possible errors in merging datasets), which
are now potentially out of date considering the scale of anthropogenic riverbed lowering in the VMD. They also do not
incorporate the potential for increased frequency of storm surges in the Bassac River region, as a consequence of projected
climate change.

Therefore, our study aims at assessing compound flooding dynamics of the coastal environs of the Ca Mau Peninsula. Our
focus lies in assessing potential shifts in the diverse factors that serve as triggers for flooding events in this region. Recognizing
the profound significance of this inquiry, particularly concerning the heightened vulnerability of local communities to flooding,
our overarching aim is to refine decision-making frameworks, enhance flood management strategies, and bolster regional
planning initiatives. Through a nuanced exploration of both natural phenomena and anthropogenic influences, we seek to
furnish nuanced insights capable of informing proactive measures aimed at mitigating flood hazards. Our ultimate aspiration
is to engender greater resilience among the populace inhabiting the Ca Mau Peninsula in the face of recurrent flood


occurrences. Therefore, this article represents an important contribution for analysis, forecasting, warning, and proposing various flood control solutions for the Ca Mau region. Based on future predictions, water control systems and future developments will be planned, including expressways, coastal embankments to control tidal surges in the West Sea, East Sea, and saltwater control gates along the Hau River. The article aims to achieve the following objectives:

1. Develop a set of flooding calculation tools for the Ca Mau Peninsula with a higher level of detail than existing tools to support long-term research and development for the region;

2. Analyse in detail for the first time the causes of compound flooding on the Ca Mau Peninsula, quantifying the impact of each cause and compound factors based on hydraulic models. The study examined both present and future scenarios, encompassing variations in upstream freshwater flow, human-induced riverbed lowering, land subsidence, and eustatic sea-level rise; and

3. Establish and compare the levels of inundation change in the study area, serving as a basis and direction for the construction and development of infrastructure to protect residents, transportation (expressways, internal routes), and the distribution of more effective industrial clusters and wider VMD zones.

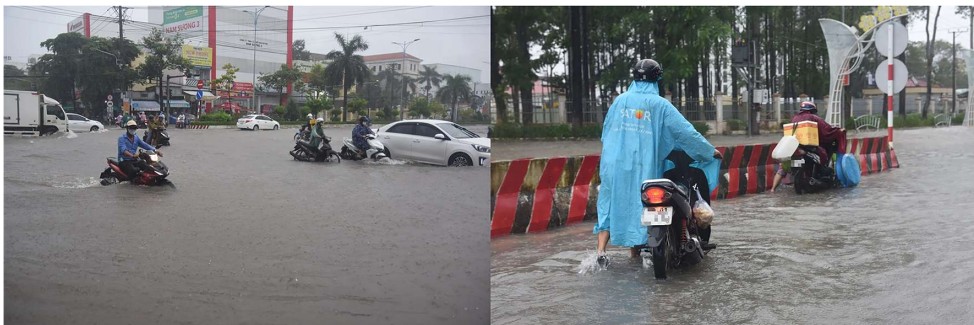

**Figure 1. The extensive flooding within Ca Mau city (see Fig. 2a for exact location), resulting from intense rainfall and elevated tides on October 2, 2023 (source: thanhnien.vn)**

## 2 Study Area, Flood Model and Methods

### 2.1 Study area

The Ca Mau Peninsula, located in the southern coastal region of the VMD, covers an area of about 5,210 km$^2$ and is current inhabited by 1.19 million people (General Statistics Office of Vietnam, 2020) (Figure. 2a). In 2020, the aquaculture sector in the Ca Mau Peninsula expanded to cover an area of 285.5 thousand hectares. This constituted 37.16% of the aquaculture area within the Mekong Delta region and 26.49% of the nation's total aquaculture area. The total fishery output reached 590,191 tons, contributing to 12.2% and 6.8% of the fishery output within the VMD and the entire nation, respectively (General Statistics Office of Vietnam, 2020). In addition, with a rich mangrove ecosystem that covers 77% of the mangrove forests in the VMD region, the Ca Mau Peninsula holds a pivotal role in overseeing, preserving, and fostering the sustainable development of the Mui Ca Mau Biosphere Reserve (Tinh et al., 2009; Son et al., 2015; Thuy et al., 2020). It stands out as





one of the nation's premier geographical, cultural, and ecological attractions, characterized by its distinctive river mouth and
coastal ecosystems.

The hydro-climate in the broader VMD region is tropical, with its hydrology heavily influenced by a monsoon seasonal flood-pulse, marked by distinct dry and flood seasons. The flood season typically spans from mid-May to October, representing over 90% of the annual precipitation in the VMD (Kingston et al., 2011), and accounts for 80-90% of the total annual river flows (Triet et al., 2017). The Ca Mau Peninsula is a low-lying area that is frequently flooded. Its average elevation ranges from 0.5

130   m to 1.5 m above mean sea level, with a coastline length of 254 km with 154 km bordering the Gulf of Thailand in the west and 100 km facing the East Sea in the east. In the East Sea, a semi-diurnal tidal pattern with an amplitude ranging from 2.5 to 4.0 m is observed, whereas the Gulf of Thailand experiences a diurnal tidal regime with amplitudes varying from 0.8 m to 1.2 m (Gugliotta et al., 2017). This region is characterized by a dense network of natural and human-made rivers, canals and is currently undergoing infrastructure development, including expressways, coastal embankments, and water control works along

the West Sea and East Sea coasts.

## 2.2 Flood model

We utilized a 1D hydrodynamic model covering the entire Mekong Delta, extending from Kratie, Cambodia to the coastal zone in Vietnam.  The model was initially developed by Dung et al., (2011) using the modelling package MIKE11-HD developed by the Danish Hydraulic Institute (DHI). It was set up to represent the complex river network and floodplains and

numerous flood control structures (e.g dikes, sluice gates) in the VMD (Fig. 2b). The primary input boundary is the daily discharge at Kratie. Tidal stage measurements from coastal stations along the VMD coastal area are utilized to define downstream boundary conditions. The model utilises around 30,000 computational nodes around the domain.


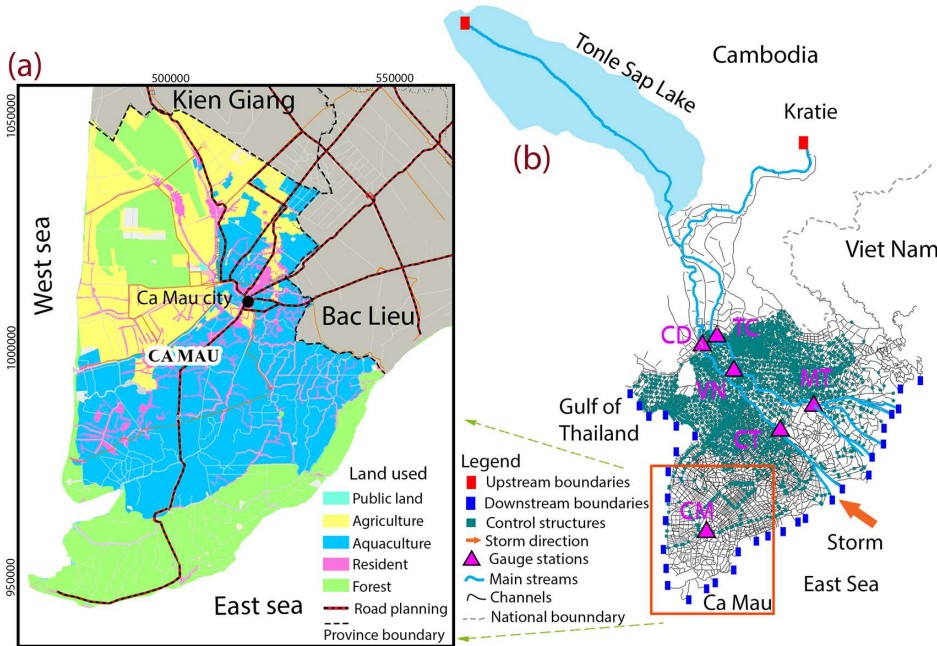

**Figure 2. (a) The study area's location and the land use for the year 2022, along with the projected road infrastructure development in the study area up to 2030; (b) the 1D model used in this study**

A key contribution of this study is the updating of datasets to be fed into the modelling. This includes updating dyke heights throughout the VMD, from data provided by Southern institute of water resource research (SIWRR) (surveys carried out 2018-2019). Additionally, the tidal and saline prevention culvert systems along the Western coastal of VMD have been updated up to 2019. Riverbed topography data for the mainstream of the Mekong River within VMD was updated in 2018, based on the study of (Vasilopoulos et al., 2021). This bathymetry data was incorporated at approximately 3-km intervals along the entire Mekong and Bassac channels. The study utilises hydrological data pertaining to water discharge and water level, gathered from various stations situated across the primary channels of the VMD (Table 1). The dataset comprises a single set of long-term daily water flow data at Katie, obtained from the Mekong River Commission (MRC). Other data, including water discharge and water level in the mainstream Mekong River, are recorded on an hourly basis. Additionally, daily maximum and minimum water levels within the study area have been collected. These measurements are obtained from gauges operated by the Southern Regional Hydro-Meteorological Center (SRHMC) and are referenced to the Hon Dau Mean Sea Level (MSL) datum. The specific locations and names of each station can be found in Table 1 and Figure 2. A Digital Elevation Model (DEM) for the Ca Mau peninsula, has been acquired from the Ministry of Natural Resources and Environment (MONRE) in 2008 for the purpose of creating inundation maps (Appendix A, Fig. A1). This DEM utilizes the WGS84 coordinate system and is based on the Hon Dau mean sea level vertical datum, featuring a resolution of 5 m x 5 m.





**Table 1. Information on Gauge Stations and data availability**

| No | Gauges (code) | Latitude ($^0$N) | Longitude ($^0$E) | Available data period | | Resolution | Channel |
|---|---|---|---|---|---|---|---|
| | | | | Discharge | Water level | | |
| 1 | Kraite (KH_014901) | 12.481 | 106.018 | 2000-2021 | 1933-2022 | Daily | Mekong |
| 2 | Tan Chau (TC) | 10.801 | 105.248 | 2016-2018 | 2016-2018 | Hourly | Mekong |
| 3 | Vam Nao (VN) | 10.417 | 105.644 | 2016-2018 | 2016-2018 | Hourly | Mekong |
| 4 | My Thuan (MT) | 10.275 | 105.926 | 2016-2018 | 2016-2018 | Hourly | Mekong |
| 5 | Chau Doc (CD) | 10.705 | 105.134 | 2016-2018 | 2016-2018 | Hourly | Bassac |
| 6 | Can Tho (CT) | 10.053 | 105.787 | 2016-2018 | 2016-2018 | Hourly | Bassac |
| 7 | Ca Mau (CM) | 09.176 | 105.155 | - | 2016-2018 | Daily | Ganh Hao |

**2.3 Model calibration and validation**

To assess the precision of the updated model, it was subjected to testing under different hydraulic upstream conditions. Specifically, the model has been calibrated using data from the high-water flux event in 2018, with a water volume of 454 billion m$^3$ at Kratie (Fig. 3). In comparison, the annual water volume at Kratie from 2000 to 2021 averaged at 390 billion m$^3$. The 2018 hydraulic year had annual volume exceedance frequency is approximately 20%, relative to the 2000-2021 data range.

Following this, the model's validation was carried out using data from the year 2016, which was linked to a low-water flux event, where the water volume amounted to 331 billion m$^3$ (Fig. 3). During this period, the annual volume exceedance frequency was approximately 78%, compared to the historical data range spanning from 2000 to 2021. The choice of these particular years was based on their close alignment with the observed bathymetry used in the model, signifying minimal changes in the riverbed. As a result, the model underwent calibration for a high flood event, and validation against a low flood

event, enabling an evaluation of the model's suitability across a broad range of potential flood magnitudes. The calibration and validation procedures involved using observed upstream water discharge values at Kratie gauges, recorded on a daily basis. Additionally, downstream boundary conditions were determined based on hourly tidal stage measurements taken along the Mekong coastal zone for the years 2018 and 2016, respectively.

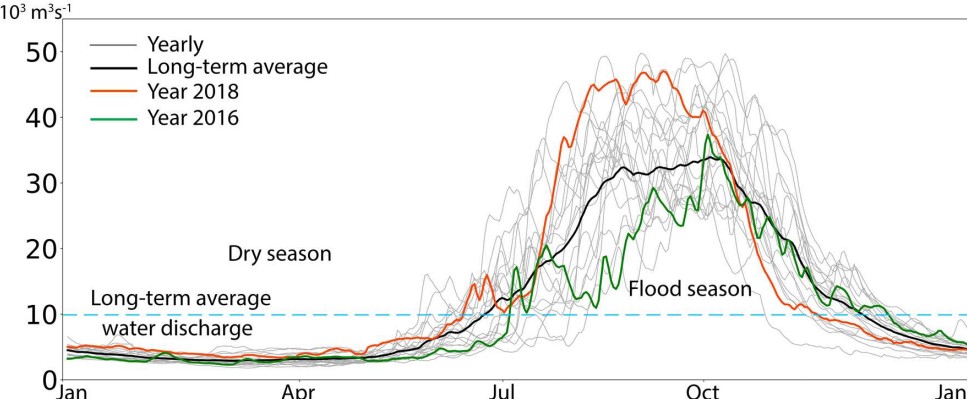

**Figure 3. The daily discharge at Kratie was utilized as the upstream boundary for the modelling scenarios, which including the daily long-term average water discharge spanning from 2000 to 2021 and the observed water discharge for the year 2018 and 2016**

The calibration procedure entailed fine-tuning of the Manning roughness parameters, a standard practice in hydraulic models. This process utilized monitoring stations for water discharge and water levels positioned along the primary Mekong channel in Vietnam (Fig. 2b and Table 1). The model's predictions were extracted and compared with the corresponding observed data. In cases where the model's performance was deemed unsatisfactory, adjustments were made to the roughness coefficient in those specific zones. The model was then rerun through multiple iterations until optimal results were achieved.

The Nash-Sutcliffe model efficiency coefficient ($NSE$), as specified in Eq. (1), was utilized to assess the accuracy of the comparison between the recently calibrated predictions and the observed data (Nash and Sutcliffe, 1970).

$$NSE = 1 - \frac{\sum_{t=1}^{T}(X_m^t - X_0^t)^2}{\sum_{t=1}^{T}(X_0^t - \overline{X_0})^2} \tag{1}$$

In this equation, $\overline{X}_0$ represents the mean of observed values, $X_m^t$ is the calibrated value at time t, and $X_0^t$ is the observed data at time $t$. Generally, $NSE$ values below 0.5 indicate suboptimal calibration performance, while $NSE$ values surpassing 0.5 suggest a satisfactory model performance. $NSE$ values exceeding 0.65 indicate a well-performing calibration, and values surpassing 0.8 indicate a highly accurate calibration (Ritter and Muñoz-carpena, 2013).

The model is run for both dry periods, characterized by water discharge falling below the long-term average values spanning from 2000 to 2021, and flood seasons, identified by days when water discharge surpasses the average values within the same dataset. These definitions align with those outlined by the MRC, (2011). In the calibration process of inundation within the Ca Mau peninsula, the deviation value is utilized to assess the accuracy between the modelled and observed data. The water level at the Ca Mau station, situated in the central part of the study area, serves as the basis for this assessment. The deviation value was also calculated, as specified in Eq. (2).

$$Dev = \left|\frac{Obs - sim}{Obs}\right| * 100\% \tag{2}$$



where *Sim* is the simulated maximum water level value and *Obs* is corresponding observed data. The best performant has a *Dev* of 0. Calibration establishes the most appropriate Manning coefficient values for the model. In the subsequent validation phase, the model was tested using data from the different hydraulic year of 2016.

In the calibration step, the results demonstrate outstanding consistency between the simulated and observed water levels and
water discharge throughout the simulation period and across the entire spatial extent of the VMD (Fig. 4a). The average *NSE* values for these measurement points exceed 0.8, with the exception of the Vam Nao station, which *NSE* for discharge calibration registered at 0.63. This suggests the effective performance of the model (Table 2). The results also emphasizes that the agreement between the predicted water levels and the corresponding observed values tends to be higher than that for water discharge across all gauging stations. While there is generally good agreement between the predicted and observed maximum
water levels, it's noteworthy that the observed maximum water level tends to be slightly higher than the corresponding value at the Can Tho station. The simulated maximum water level at the Ca Mau gaging station was extracted and compared with the observed data. The results indicate that, during the calibration phase, the simulated maximum water level was approximately 0.92 m, compared to the observed value of 0.84 m, resulting in a deviation (*Dev*) of 10%, indicating a satisfactory level of agreement.



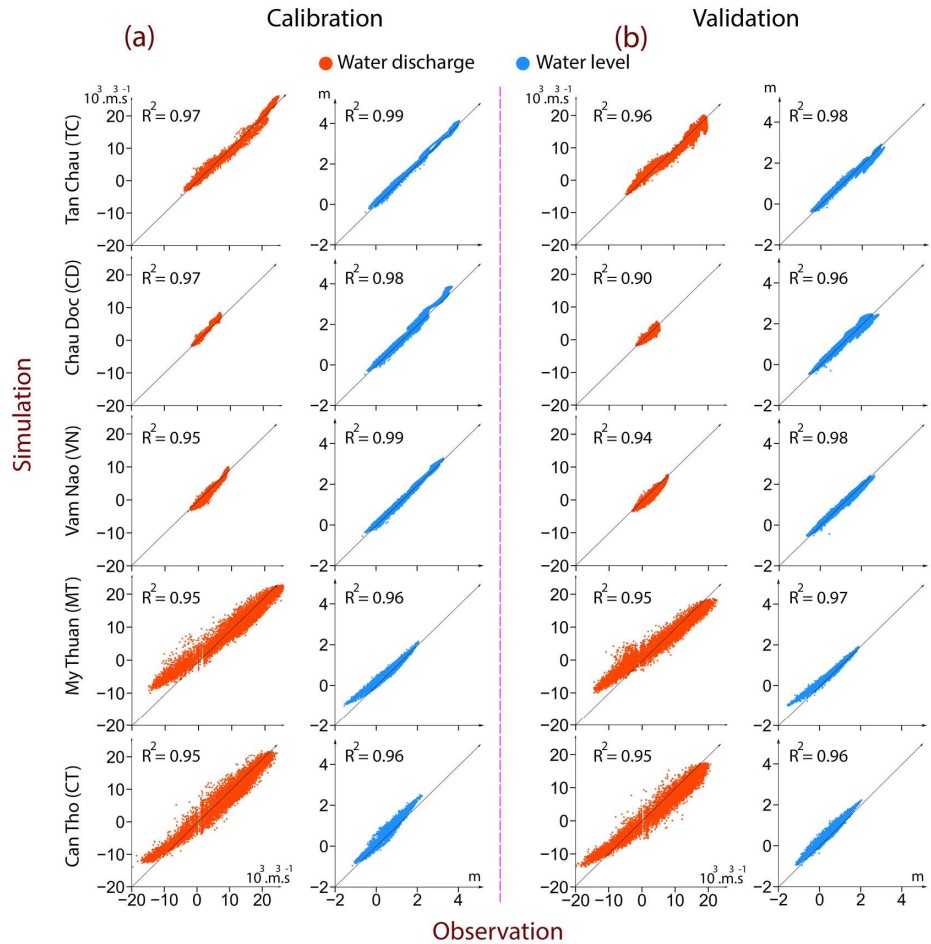

**Figure 4.** Assessing the agreement between the predicted and observed hourly data for water level and water discharge at monitoring stations throughout the VMD during the calibration run in 2018 and validation run in 2016.

**Table 2.** The *NSE* coefficient was calculated across a range of river gauge stations spanning the VMD, averaging the values for distinct dry and flood season periods during the calibration step

| No | Water level (WL) | | Water discharge (Q) | |
|---|---|---|---|---|
| | Dry | Flood | Dry | Flood |
| Tan Chau | 0.90 | 0.97 | 0.94 | 0.93 |
| Chau Doc | 0.96 | 0.94 | 0.92 | 0.93 |




| | | | | |
|---|---|---|---|---|
| Vam Nao | 0.97 | 0.97 | 0.63 | 0.83 |
| My Thuan | 0.98 | 0.97 | 0.93 | 0.93 |
| Can Tho | 0.88 | 0.88 | 0.95 | 0.94 |

For the validation stage, the results reveal a consistently strong agreement between the simulated outcomes and the corresponding observed data across the VMD gauging stations (Fig. 4b and Table 3). There is a persistent pattern of high agreement in water level values and an overall strong agreement in terms of water discharge. At the Ca Mau gauging station, the simulated maximum water level was 0.86 m, while the observed value was 0.77 m, yielding a *Dev* of 12%. This deviation implies a reasonably good agreement between the simulated values and the observed data.

**Table 3. The *NSE* coefficient was calculated across a range of gauge stations spanning the VMD, averaging the values for distinct dry and flood season periods during the validation step**

| No | Water level (WL) | | Water discharge (Q) | |
|---|---|---|---|---|
| | Dry | Flood | Dry | Flood |
| Tan Chau | 0.95 | 0.98 | 0.91 | 0.87 |
| Chau Doc | 0.98 | 0.94 | 0.87 | 0.79 |
| Vam Nao | 0.98 | 0.97 | 0.73 | 0.68 |
| My Thuan | 0.98 | 0.97 | 0.93 | 0.91 |
| Can Tho | 0.91 | 0.83 | 0.94 | 0.92 |

The calibration and validation results reveal that the updated flood model is performing well in capturing flood dynamics across the VMD. By integrating 2018 riverbed bathymetry data and 2019 infrastructure information, particularly in the main river system, we've significantly improved the model's accuracy. Calibration for the high-water year of 2018 and validation for the low-water volume year of 2016 have yielded highly reliable results, showcasing the model's ability to simulate diverse hydraulic conditions effectively. Consequently, this model will be utilized to comprehensively investigate the flooding regime in the VMD through various simulation scenarios.

**2.4 Simulation Scenarios**

In order to assess the potential future inundation across the study area, a range of scenarios involving variations in upstream freshwater flow, human-induced riverbed lowering, land subsidence, and eustatic sea-level rise were introduced into the modelling process for analysis. The details of each scenario are outlined in Table 4 and are discussed in more detail in the sub-sections below.





### 2.4.1 Individual drivers

First, we ran a Baseline Scenario (S0). This simulation uses the daily time series of long-term mean flow at Kratie ($Q_M$) spanning from 2000 to 2021 (Fig. 3), where the water volume at Kratie is approximately 390 billion m$^3$ and the downstream boundary conditions were determined using hourly tidal stage measurements taken along the Mekong coastal zone for the year 2018. This scenario serves as the foundation for modelling average flooding in the Ca Mau Peninsula and is employed for comparison with alternative flood scenarios.

We then ran 7 other scenarios, varying the upstream discharge, storm surge, lowering the riverbed, accounting for land subsidence and sea level rise, as described below.

**High upstream flow (S1):** The observed water discharge at Kratie for the year 2018 ($Q_{20\%}$) was employed as the upstream water discharge, which involved a water volume estimated at around 454 billion m$^3$ and the total volume exceedance frequency is approximately 20% within the historical data range from 2000 to 2021. This scenario is utilized to evaluate inundation in the study areas under high water upstream flux.

**Storm surge (S2):** This scenario assesses the impact of storm surge inundation, caused by tropical cyclone, in the Ca Mau Peninsula. The chosen location for the scenario's storm surge impact is the Bassac channel mouth along the VMD coastal area. This selection is based on the significance of the Bassac River as the primary stream in the VMD, capable of transferring the storm surge impact further inland. Additionally, this channel is in close proximity to the study area. Information on the tropical cyclone annual exceedance probability of 1% AEP ($TC_{1\%}$) affecting the VMD at the Bassac River mouths, has been utilized for this scenario. The timing of the tropical cyclone occurrence was chosen in the middle of September, coinciding with the period when flooding caused by high upstream water discharge typically takes place. The rise in water level due to the tropical cyclone impacting the VMD at the Bassac River mouth has been synchronized with a full tidal cycle in the VMD coastal zone at the time of the storm, serving as the downstream boundaries. This integration results in the maximum water level at the Bassac River mouth boundary rising up to 2.9 m during the storm. These storm surge data for the VMD coastal zone has been adopted from Wood et al., (2023).

**Human-induced riverbed lowering (S3):** This scenario evaluates the effects of riverbed lowering caused by upstream damming and sand mining within the VMD delta on inundation in the study area. The anticipated bathymetry for the main Mekong channel within the VMD in 2038 ($B_{2038}$) as derived from Vasilopoulos et al., (2021) has been employed in the scenarios. This riverbed bathymetry for 2038 was generated by incorporating the riverbed bathymetry from 2018 into the riverbed lowering trend observed from 2008 to 2018, as detailed in Vasilopoulos et al., (2021). This results in the riverbed bathymetry for 2038 being lower than that of 2018, with an average decrease of 2.8 m. The cross-sections, spaced approximately every 3 km along the entire Mekong and Bassac channels, for the 2038 bathymetry were then integrated into the model.

**Delta Subsidence (S4_a, S4_b):** These scenarios evaluated the inundation change associated with land subsidence in the future. The spatial map depicting land subsidence within the VMD in the projected future is derived from Minderhoud et al.,





(2020). This dataset, encompassing land subsidence scenarios labelled as B1.5 for the year 2050 ($B1.5_{2050}$) and B2 for the year 2100 ($B2_{2100}$) in the study by Minderhoud et al., (2020), is denoted in this study as S4_a and S4_b scenarios, respectively.

The S4_a and S4_B scenarios involve the average land subsidence around 0.38 m (σ =0.21 m) and 1.12 m (σ =0.59 m) in the Ca Mau peninsula, respectively.

**Sea-level rise (S5_a, S5_b):** These scenarios evaluated SLR, an inevitable consequence expected to persist for centuries to millennia due to ongoing deep ocean warming and ice sheet melting. However, human activities, particularly the release of greenhouse gases, have unequivocally contributed to global warming (IPCC, 2023). One of the outcomes of this trend is rising

sea levels, extensively explored in various studies (e.g., Kopp et al., 2014; Garner et al., 2018; IPCC, 2023). Since SLR is not just a future prediction but a current phenomenon impacting coastal regions worldwide, it requires attention not only for long-term strategic planning but also for immediate emergency readiness and other short-term considerations (Hall et al., 2019). In this study, consistent with the forecasts outlined in the IPCC Sixth Assessment Report (AR6) (NASA, 2021) for the Mekong Delta coastal zone, two scenarios for eustatic SLR were developed. These scenarios involve projections of 0.5 m for the mid-

century (2050) (referred to as S5_a) and 1.0 m for the end of this century (2100) (referred to as S5_b), under SSP5-8.5 scenarios with low confidence, in comparison with the sea level in the year 2018, respectively. Following this, the values representing sea-level rise scenarios are incorporated into the time series of tidal levels from the year 2018, functioning as downstream boundaries for the model scenarios.

### 2.4.2 Multiple/combined drivers

We then ran 2 other scenarios in which we combined drivers, as described below.

**Projected until Mid-Century (S6):** This scenario combined inundation changes when the high upstream flow (observed water flux of 2018) with storm of 1% AEP entering the Bassac River mouth, incorporating lowered riverbed (2038 riverbed bathymetry), subsidence scenario until 2050 and SLR until 2050.

**Projected until End of the Century (S7):** This scenario combined inundation changes when the high upstream flow (observed

water flux of 2018) with storm of 1% AEP entering the Bassac River mouth. This scenario incorporates the lowered riverbed (2038 bathymetry), the severe subsidence scenario until the end of the century (B2_2100 scenario), and SLR until 2100.

**Table 4. Scenarios utilized to explore the impact of elevated upstream water flow, storm surge, anthropogenic riverbed lowering, delta subsidence, and sea-level rise on the risk of inundation for the Ca Mau peninsula.**

| Scenarios | Upstream discharge | Storm surge | Riverbed lowering | Delta land subsidence | Sea level rise |
|---|---|---|---|---|---|
| **S0 (baseline)** | $Q_M$ | | | | |
| **Scenarios based on individual drivers** | | | | | |




| S1 | $Q_{20\%}$ | | | | |
|---|---|---|---|---|---|
| S2 | $Q_M$ | $TC_{1\% \, AEP}$ | | | |
| S3 | $Q_M$ | | $B_{2038}$ | | |
| S4_a | $Q_M$ | | | $B1.5_{2050}$ | |
| S4_b | $Q_M$ | | | $B2_{2100}$ | |
| S5_a | $Q_M$ | | | | + 0.5 m |
| S5_b | $Q_M$ | | | | + 1.0 m |
| **Scenarios based on individual drivers** | | | | | |
| S6 | $Q_M$ | $TC_{1\% \, AEP}$ | $B_{2038}$ | $B1.5_{2050}$ | 0.5 m |
| S7 | $Q_M$ | $TC_{1\% \, AEP}$ | $B_{2038}$ | $B2_{2100}$ | 1.0 m |

**2.5 Inundation mapping and statistics**

The direct outcome of the 1D flood model includes hourly water levels at all computational nodes (~30,000) within the VMD, for a simulation period 1st June to 30th November for each year simulated. Subsequently, we extracted the maximum water level and utilized it for spatial interpolation (using the Natural Neighbour method) to create a water level map with a resolution of 5 m across the study area. The maximum inundation map was then derived by subtracting the ground elevation from the DEM of region. Areas with a depth <0.1 m are classified as unflooded. These post-processing steps were automated using

ArcGIS. Flood maps illustrating the change in inundation levels between simulation scenarios (see Section 2.4) have also been derived. Based on this mapping, we estimated flooding area statistics, providing flooding areas corresponding to each of the following 10 classes: below 0.1 m, 0.1 m – 0.4 m, 0.4 m – 0.7 m, 0.7 m – 1.0 m, 1. 0 – 1.3 m, 1.3 m – 1.6 m, 1.6 m – 1.9 m, 1.9 m – 2.2 m, 2.2 – 2.5 m, above 2.5 m. Additionally, maps illustrating the increasing risk of flooding by comparing different scenarios with the S0 scenario have been created. We estimated increasing flooding area statistics for the 10 inundation classes:

below -0.1 m (reduce the inundation), -0.1 m – 0.1 m (unchanged), 0.1 m – 0.4 m, 0.4 m – 0.7 m, 0.7 m – 1.0 m, 1. 0 – 1.3 m, 1.3 m – 1.6 m, 1.6 m – 1.9 m, 1.9 m – 2.2 m, above 2.2 m.



## 3 Results

Spatial maps depicting the maximum water level, inundation, and the inundation change, relative to the baseline, are shown in Fig. 5, 6 and 7, respectively, for each scenario in the Ca Mau Peninsula. The key results for each of the scenarios are described

below. The areas flooded in the Ca Mau Peninsula region under the scenarios are listed in Table 5. The accumulated increase in flooded area, compared to the baseline, is given in Table 6.

In the baseline scenarios (S0), the average maximum water level in the study area could reach 0.81 m (σ =0.44 m), and the peak of the maximum water level could reach 1.8 m in the East Sea coastal zone (Fig. 5). This scenario could result in an average inundation depth of 0.04 m (σ =0.43 m) (Fig. 6). Furthermore, within these baseline scenarios, the estimated flood

coverage is about 23.9% of the study area, featuring inundation depths between 0.1 m and 0.4 m (Table 5).

In scenario S1 (which evaluates the impact of high upstream water discharge involved the total annual volume exceedance frequency of approximately 20% within the historical data range from 2000 to 2021), the average maximum water level shows a slight increase, reaching 0.92 m (σ =0.45 m), and the peak of maximum water level could reach 1.92 m in the East Sea coastal zone (Fig. 5). This highlights the small rise in water levels in this coastal area resulting from the combination of high-water

discharge and sea water level. Moreover, this scenario results in a rise in average inundation depth to 0.15 m (σ =0.43 m) (Fig. 6), affecting around 27.8% of the area with flood depths between 0.1 m to 0.4 m and nearly 16.2% of study area with inundation depths ranging from 0.4 m to 0.7 m (Table 5). The impact of solely high upstream water discharge results in a 43% increase in the study area where inundation depths range from 0.1 m to 0.4 m compared to the corresponding values in scenario S0 (Fig. 7 and Table 6).

In scenario S2 (which evaluates the effect of a storm surge with exceedance probability of 1% AEP reaching the Bassac river mouth; Fig. 2b), the results indicate minimal changes in both the maximum water level and the extent of inundation compared to the baseline scenarios S0 (Figures 5-7, Table 5-6). This implies that the potential impact of a storm surge reaching the main Mekong channel is likely to be minimal in this area. Similarly, in scenario S3, where the effect of riverbed lowering on the main stream Mekong and Bassac channel is assessed, the findings also demonstrate insignificant alterations in both the

maximum water level and the extent of inundation when compared to the baseline scenarios S0 (Figures 3-5, Table 3).

In scenario S4_a (which investigates the effects of land subsidence up to 2050), results underscore the importance of land subsidence on area inundation. Specifically, the average inundation level shows an increase, rising from 0.04 m (σ =0.43 m) in the S0 scenarios to 0.42 m (σ =0.44 m). The peak of inundation level could reach 3.93 m in the vicinity of Ca Mau city (Fig. 6). In this scenario, it is projected that the inundated area would cover approximately 30.8% of the study area, with flood

depths ranging from 0.1 m to 0.4 m. Furthermore, this land subsidence scenarios could increase the flooding risk by an average of 0.38 m (σ =0.21 m), with a maximum increase in flooding risk of 0.94 m compared to that in the base scenarios S0 (Fig. 7). More specifically, the effects of land subsidence projected until 2050 could lead to an expansion of 43.3% of the study area, experiencing an increase in inundation depths ranging from 0.1 m to 0.4 m compared to the baseline scenarios S0. Additionally,





approximately 20.3% and 9.1% of the study area could see rises in inundation depths ranging from 0.4 m to 0.7 m and 0.7 m

to 1.0 m, respectively, compared to the baseline scenarios S0 (Table 6).

In scenario S4_b (which analyses the impact of land subsidence up to the end of this century), the results demonstrate a noticeable increase in inundation both spatially and in magnitude. Specifically, the projected land subsidence by 2100 could elevate the average inundation depth to 1.15 m (σ =0.67 m) (Fig. 6). This would encompass an estimated 7.9% of the study area, with inundation depths ranging from 0.1 m to 0.4 m, while approximately 17.6% and 19.8% of the area would experience

depths ranging from 0.4 m to 0.7 m, and from 0.7 m to 1.0 m, respectively. Moreover, 1.9% of the study area is expected to encounter inundation depths exceeding 2.5 m (Table 5). Consequently, this land subsidence level could increase the flooding by an average of 1.12 m (σ =0.59 m), with the maximum increase in flooding risk being 2.44 m compared to those in the base scenarios S0 (Fig. 7). The effects of land subsidence up to 2100 may lead to a 22.8 % expansion of the area, where inundation depths increase ranging from 0.4 m to 0.7 m compared to those in the base scenarios S0. Additionally, approximately 6.4% of

the study area, could experience inundation depths exceeding 2.2 m in comparison with the base scenarios S0 (Table 6).

In scenario S5_a (which evaluates the impact of a SLR up to 2050 with a value of 0.5 m), the results indicate that the increase in sea level could elevate the average maximum water level up to 1.32 m (σ =0.45 m), with the maximum water level reaching 2.30 m in the East Sea coastal zone (Fig. 5). Moreover, this rise in sea level also results in an increase in the average inundation depth by 0.55 m (σ =0.43 m) (Fig. 6), affecting approximately 23.9% of the study area with inundation depths ranging from

0.1 m to 0.4 m (Table 5). This scenario exacerbates the risk of inundation at higher levels, impacting around 29.9% of the study area with inundation depths ranging from 0.4 m to 0.7 m and nearly 20.4% of the study area with inundation depths ranging from 0.7 m to 1.0 m (Table 5). Regarding the comparison with corresponding values in S0 scenarios, it is highlighted that the sea level rise of 0.5 m could increase both the average and peak of maximum water level to 0.51 m (σ =0.03 m) and 0.62 m respectively, in comparison with the corresponding values in S0 scenarios (Fig. 7). Furthermore, the influence of a sea

level rise of 0.5 m could lead to an expansion in the 86.7% of study area, where the inundation depths increase ranging from 0.4 m to 0.7 m compared to those in the S0 scenarios (Table 6).

In scenario S5_b (which examines the impact of a 1.0 m sea level rise by 2100), the findings reveal that this increase in sea level elevates the average maximum water level and peak of maximum water level to 1.83 m (σ =0.44 m) and 2.80 m respectively. Consequently, almost the entire study area is inundated, with an average inundation depth of 1.06 m (σ =0.43 m)

and a peak maximum inundation depth of 4.45 m. In this scenario, approximately 28.7% of the study area experiences inundation depths ranging from 0.7 m to 1.0 m. However, it exacerbates the risk of inundation at higher levels, affecting around 28.0% of the study area with inundation depths ranging from 1.0 m to 1.3 m (Table 5). Regarding the increase in inundation compared to the corresponding values in S0 scenarios, it is observed that a sea level rise of 1.0 m could elevate both the average and peak of the maximum water level to 1.02 m (σ =0.03 m) and 1.20 m, respectively, compared to the corresponding values

in S0 scenarios (Fig. 7). Furthermore, the impact of a SLR of 1.0 m could lead to an expansion in the area comprising 78.3% of the study area, where inundation depths increase ranging from 0.7 m to 1.0 m, and 19.4% of the study area, where inundation depths increase ranging from 1.0 m to 1.3 m, compared to those in the S0 scenarios (Table 6).





In scenarios S6 (which combine a series of drivers up to 2050; Table 4), the combined factors drive the average maximum water level up to 1.33 m (σ =0.43 m), with the peak of maximum water level reaching 2.31 m in the East Sea coastal zone

(Fig. 5). Additionally, these combined factors result in an average inundation depth of 0.91 m (σ =0.44 m), with the maximum inundation depth reaching up to 4.39 m (Fig. 6). Most areas experiencing inundation depths ranging from 0.7 m to 1.0 m cover 30.0% study area, with nearly 0.2% of study area experiencing inundation depths higher than 2.5 m (Table 5). It is noteworthy that the areas experiencing high inundation depths are in the vicinity of Ca Mau city. Regarding the increase in inundation compared to corresponding values in S0 scenarios, it is highlighted that these combined factors could elevate the average and

peak of maximum water level to 0.88 m (σ =0.22 m) and 1.38 m respectively, in comparison to the corresponding values in S0 scenarios (Fig. 7). Furthermore, the influence of these combined factors could lead to an expansion in the area of 38.8% of study area where the increasing inundation depths range from 0.7 m to 1.0 m, while there are 3.4% study area where the increasing inundation depths range from 1.3 m to 1.6 m compared to those in the S0 scenarios. (For further details, see Table 6).

In scenarios S7 (which combines a series of drivers up to 2100; Table 4), these combined factors could result in substantial inundation in the Ca Mau peninsula, driving the average maximum water level up to 1.85 m (σ =0.43 m), with the peak of maximum water level reaching up to 2.81 m in the East Sea coastal zone (Fig. 5). Additionally, these combined factors could lead to an average inundation depth of 2.20 m (σ =0.68 m) (Fig. 6). Most areas experiencing inundation depths higher than 2.5 m could cover 31.3% of study area (Table 5). It is notable that the areas experiencing high inundation depths reaching up to

6.12 m are in the vicinity of Ca Mau city.

Regarding the increase in inundation compared to corresponding values in S0 scenarios, it is highlighted that these combined factors could elevate the average and peak of maximum water level to 2.16 m (σ =0.62 m) and 3.48 m respectively, in comparison to the corresponding values in S0 scenarios (Fig. 7). Furthermore, the influence of these combined factors could lead to an expansion in the area of 2.6% of study area with the increasing inundation depths ranges from 1.0 m to 1.3 m, while

there are nearly 40.4% of study area where the increasing inundation depths exceed 2.2 m compared to those in the S0 scenarios. For further details, please refer to Table 6.

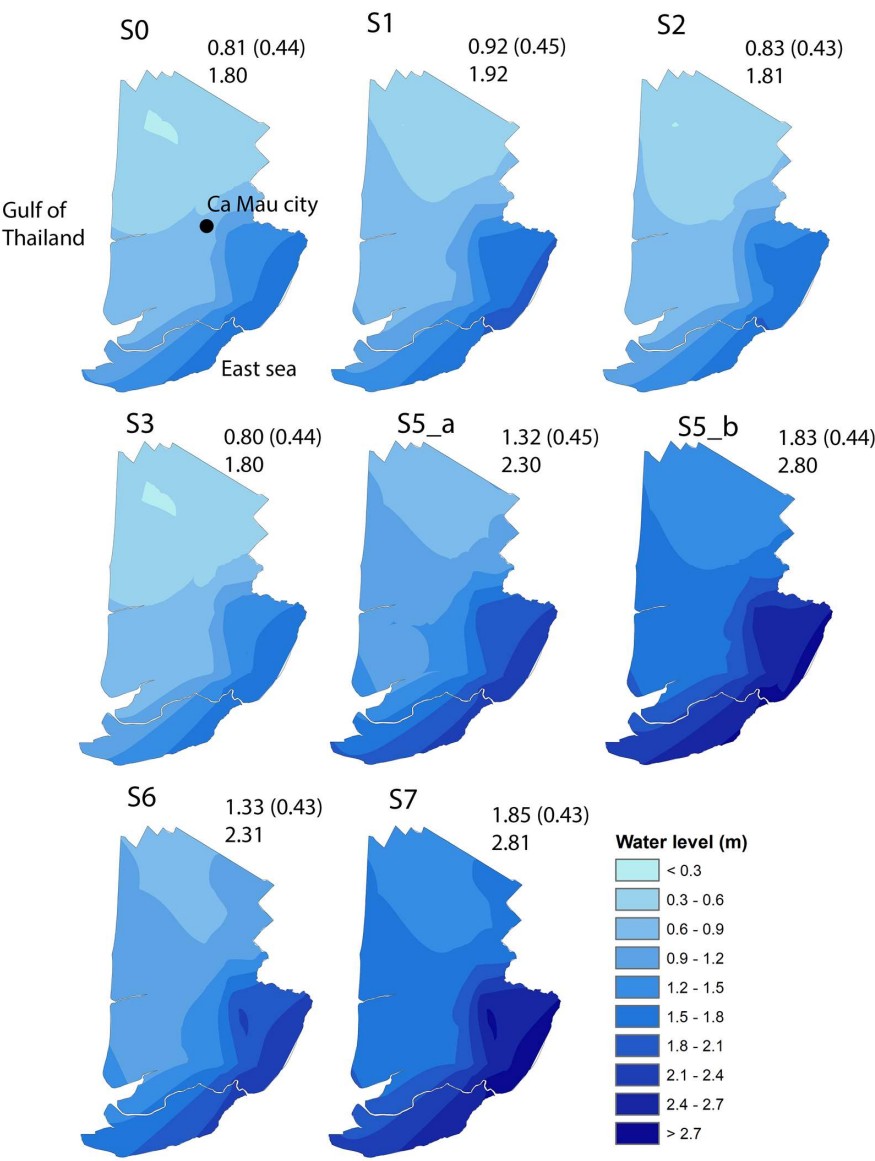

**Figure 5. The map depicting the maximum water levels in the Ca Mau Peninsula under various scenarios. The number in each panel present the average maximum water level (m) and standard deviation (m) (top), along with the peak of these maximum water levels**
**(bottom).**



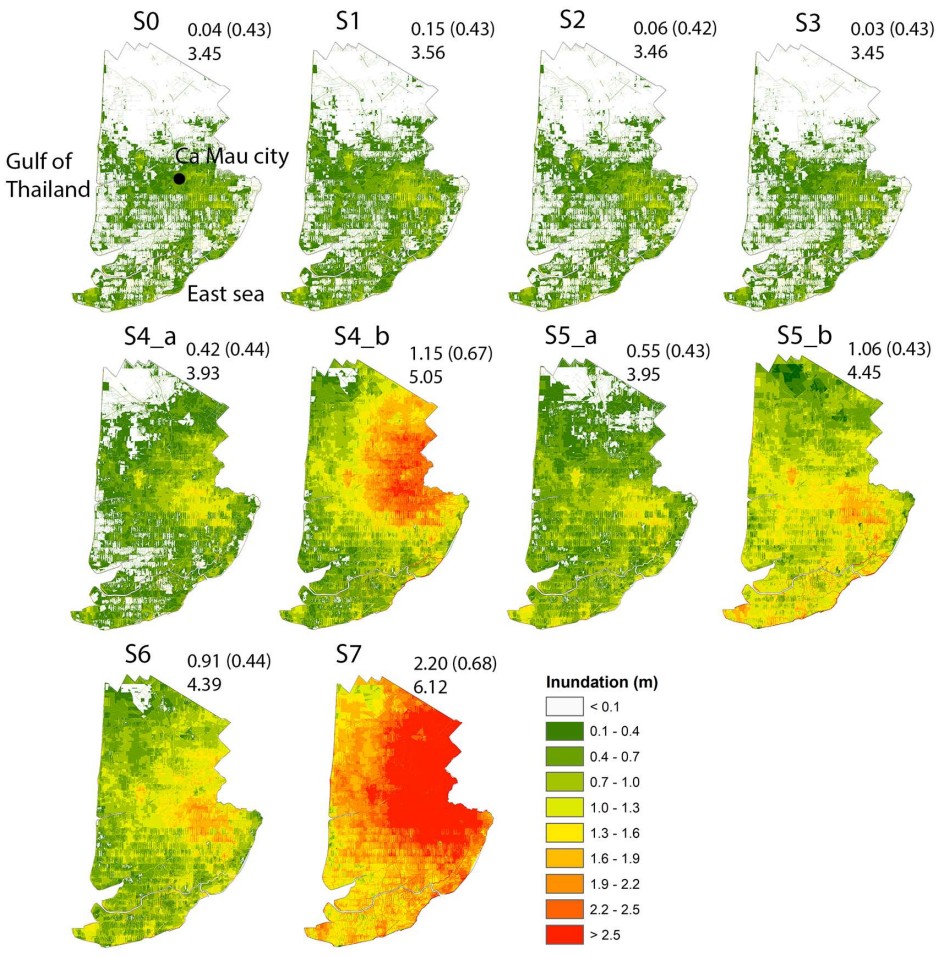

**Figure 6. The map illustrating the risk of inundation in the Ca Mau Peninsula across different scenarios. The number in each panel present the mean inundation (m) and standard deviation (m) (top), along with the peak of these inundation level (bottom).**




**Table 5. The statistics on the flooded area at different levels in the Ca Mau Peninsula region in our model scenarios**

| Level (m) | Cumulative flooded area (%) | | | | | | | | | |
|---|---|---|---|---|---|---|---|---|---|---|
| | S0 | S1 | S2 | S3 | S4_a | S4_b | S5_a | S5_b | S6 | S7 |
| 0.1- 0.4 | 23.9 | 27.8 | 25 | 23.7 | **30.8** | 7.9 | 23.9 | 3.2 | 5.8 | 0 |
| 0.4- 0.7 | 11.9 | 16.2 | 12.2 | 11.8 | 22.8 | **17.6** | **29.9** | 13.3 | 23.4 | 0.1 |
| 0.7- 1.0 | 3.2 | 5.9 | 3.8 | 3.2 | 12.6 | **19.8** | **20.4** | **28.7** | **30.0** | 1.2 |
| 1.0- 1.3 | 1.0 | 1.7 | 1.2 | 1.0 | 6.0 | 15.3 | 8.4 | **28.0** | 20.2 | 5.0 |
| 1.3- 1.6 | 0.3 | 0.6 | 0.4 | 0.3 | 1.7 | 9.8 | 2.6 | 15.8 | 10.1 | 13.6 |
| 1.6- 1.9 | 0.2 | 0.2 | 0.2 | 0.2 | 0.5 | 9.9 | 0.8 | 6.0 | 4.7 | 19.3 |
| 1.9- 2.2 | 0.1 | 0.2 | 0.1 | 0.1 | 0.2 | 7.4 | 0.3 | 1.9 | 1.4 | 17.4 |
| 2.2- 2.5 | 0.1 | 0.1 | 0.1 | 0.1 | 0.1 | 6.5 | 0.2 | 0.6 | 0.5 | 11.2 |
| >2.5 | 0 | 0 | 0 | 0 | 0.1 | 1.9 | 0.1 | 0.3 | 0.2 | **31.3** |


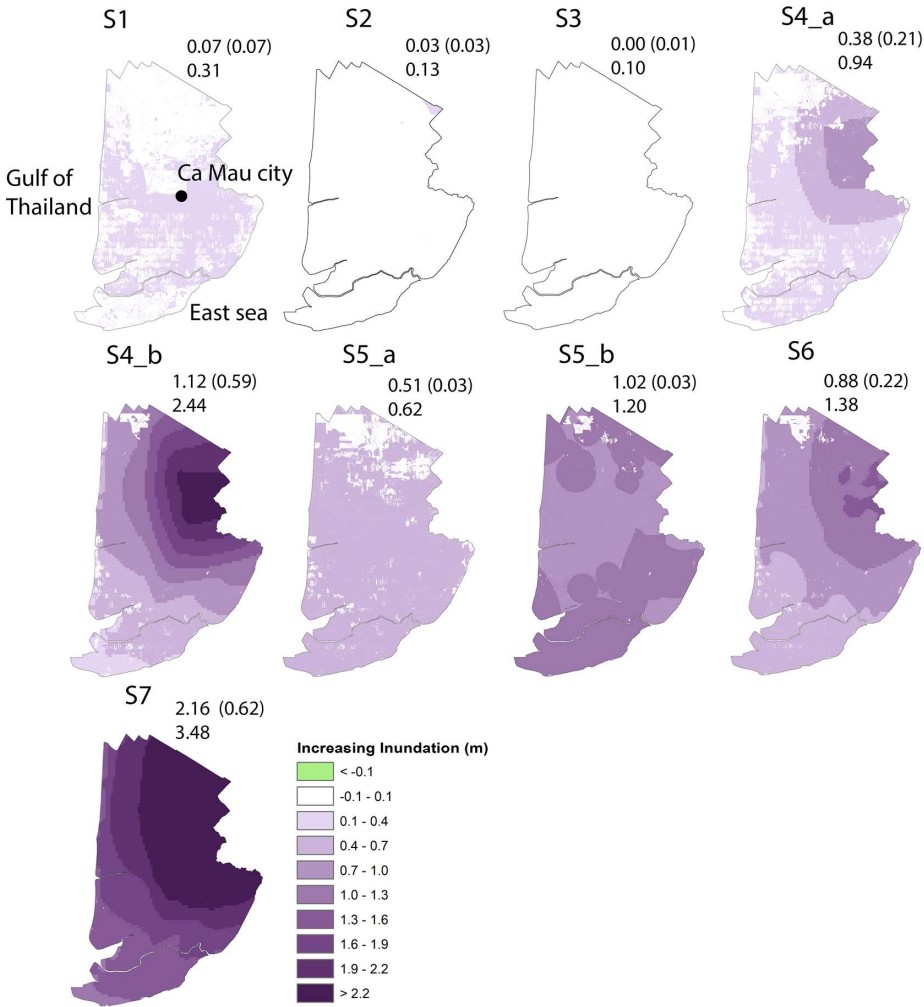

**Figure 7. The map illustrating the increasing risk of inundation in the Ca Mau Peninsula across different scenarios in comparison with the baseline scenarios S0. The number in each panel present the mean increasing of inundation level (m) and standard deviation (m) (top), along with the peak of these increasing of inundation level (bottom).**



**Table 6. The percentage increase in area of flooding in each model scenario, compared to the baseline scenario at different levels in the Ca Mau Peninsula region**

| Level (m) | Accumulated increase in flooded area (%) | | | | | | | | |
|---|---|---|---|---|---|---|---|---|---|
| | S1 | S2 | S3 | S4_a | S4_b | S5_a | S5_b | S6 | S7 |
| 0.1- 0.4 | 43.0 | 0.4 | 0 | **43.3** | 5.5 | 0 | 0 | 0 | 0 |
| 0.4- 0.7 | 0 | 0 | 0 | **20.3** | **22.8** | **86.7** | 0 | 28.8 | 0 |
| 0.7- 1.0 | 0 | 0 | 0 | 9.1 | 21.0 | 0 | **78.3** | **38.8** | 0 |
| 1.0- 1.3 | 0 | 0 | 0 | 0 | 13.5 | 0 | **19.4** | 25.3 | 2.6 |
| 1.3- 1.6 | 0 | 0 | 0 | 0 | 9.3 | 0 | 0 | 3.4 | 20.1 |
| 1.6- 1.9 | 0 | 0 | 0 | 0 | 8.7 | 0 | 0 | 0 | 19.1 |
| 1.9- 2.2 | 0 | 0 | 0 | 0 | 8.7 | 0 | 0 | 0 | 18.9 |
| >2.2 | 0 | 0 | 0 | 0 | 6.4 | 0 | 0 | 0 | **40.4** |

## 4 Discussion

The aforementioned findings underscore the potential anthropogenic impact on the evolution of the inundation regime across the Ca Mau peninsula. These results suggest that the influences of riverbed lowering in the primary Mekong channel, and the occurrence of storm surges along the mainstream Mekong River are unlikely to significantly affect the inundation regime in this region, while high-water discharge did cause the inundation in this study area to rise only by approximately 0.1-0.3 m. The minimal impacts can be attributed to the prominence of high-water discharge, riverbed lowering, and storm surges primarily occurring in the upstream region and in the main Mekong and Bassac channels. Since the study area is situated farther from these channels and closer to the sea on both sides, these impacts tend to diminish. However, the results emphasizes that land subsidence and tidal influence combined with SLR will be the primary factors driving the escalation of the inundation regime in the Ca Mau peninsula in the future, both in extent and magnitude. In the projected future by the mid-century, under the S6 scenarios, a combination of high-water discharge, storm surge, land subsidence, and sea level rise are projected to inundate 96% of the Camau Peninsula, with an average inundation depth of 0.91 m (σ = 0.44 m). Looking towards the end of the century under the S7 scenarios, the compounded effects of high-water discharge, storm surge, land subsidence, and a sea level rise are expected to inundate 99% of the Camau Peninsula, with an average inundation depth of 2.20 m (σ = 0.68 m). However, it should be noted that these projections are based on the SSP5-8.5 scenarios, which represent the very worst case with low confidence. The ramifications of this increasing trend in inundation are now deliberated in relation to the region's future sustainability and its capacity to adapt to present and future transformations.

The rising inundation resulting from land subsidence and SLR has the potential to cause saltwater intrusion into the inland regions, impacting agriculture, aquaculture, and infrastructure in the Ca Mau peninsula. Hence, there's a crucial need to guide spatial development towards creating open areas for water retention and establishing freshwater ecological zones within saline environments to promote sustainable groundwater replenishment for agriculture and aquaculture in the future. Furthermore,



it's vital to build and upgrade tidal barrier systems, taking into account the escalating inundation factors in the region, to mitigate the impact of future tidal inundation triggered by rising sea levels.

The findings highlight that Ca Mau city's urban zone (Fig. 2), characterized by its dense population, is most susceptible to flooding, underscoring the urgent need for future flood mitigation strategies. These may include measures like minimizing

surface water runoff in urban areas and promoting freshwater use for household needs. Proactive management, prevention, and control of flooding induced by SLR are imperative to guarantee the sustainable growth of coastal and riverside socio-economic zones. This effort contributes significantly to ensuring security, defence, and the protection of lives, property, and livelihoods.

Furthermore, the results highlight the potential inundation risk in this area concerning road construction standards. For

example, according to the design specifications of the national North-South Expressway for the period 2021-2025 (No: 912/QĐ-BGTVT) in the Ca Mau area, the road's crest is set at 2.2 m above mean sea level in Ca Mau city. However, in scenarios involving land subsidence of approximately 0.77 m and 2.34 m in this area by the middle and end of this century, respectively, the potential crest of the road would decrease to 1.43 m and -0.14 m by the middle and end of this century, respectively. These values may not ensure safety, especially considering the predicted maximum water levels in this area,

which are approximately 1.59 m in 2050 and 2.20 m in 2100, respectively. This assessment is based on Vietnam's road construction standards (TCVN 4054: 2005).

Moreover, the Ca Mau peninsula remains one of the areas with the richest natural ecosystem in the VMD, boasting over 79,100 hectares of mangrove forests, which contribute to its unique ecological system. In recent years, the development of transportation infrastructure, irrigation systems, and water control structures has been significantly impacting this ecosystem,

leading to its decline (Son et al., 2015; Hauser et al. 2020; Thuy et al., 2023). Combined with the potential increase in inundation in these areas, it could further exacerbate the degradation of this ecosystem. Therefore, the effective protective measures are necessary to safeguard this ecological system.

**5 Conclusions**

The Ca Mau peninsula is identified as one of the most promising economic hubs for agriculture and aquaculture within both

the Mekong Delta region and Vietnam as a whole. Nonetheless, these areas also face considerable vulnerability to flooding. This study focused on predicting the inundation patterns across the Ca Mau peninsula under different drivers using a modelling approach. The 1D hydrodynamic model, initially proposed by Dung et al., (2011), has been enhanced with updated with bathymetry data of the main Mekong River bed within the VMD in 2018, and with infrastructure features (such as dykes, sluices) in 2019 across the entire VMD. The model has been fully calibrated and validated to accurately simulate the flood

regime within the study area.

The findings indicate that factors such as increased high-water flow from upstream, alterations in the riverbed of the main Mekong channel, and occurrences of storm surges along the mainstream Mekong River are unlikely to significantly impact



the inundation dynamics in this southern region of the VMD. However, the study underscores that land subsidence, rising sea levels, and their combined effects are primary drivers behind the escalation of inundation events in the Ca Mau peninsula, both

in terms of extent and intensity, in the foreseeable future. A projected SLR of 0.5 m by the year 2050 is anticipated to result in an 87% inundation of the study area, with an average depth of 0.55 m ($\sigma$ =0.43 m). Looking ahead to the end of the century, where the estimated SLR could reach around 1 m, it is forecasted that 98% of the study area will experience inundation, with an average depth of 1.83 m ($\sigma$ =0.44 m). Furthermore, in the projected future by the mid-century, a compounded effect of factors including high-water discharge (with a total annual volume exceedance frequency of approximately 20%), storm surge

(with an annual exceedance probability of 1% AEP along the Bassac river mouth), land subsidence (averaging around 0.38 m ($\sigma$ =0.21 m), and a SLR of 0.5 m is expected to result in the inundation of 96% of the Camau peninsula area, with an average inundation depth of 0.91 m ($\sigma$ =0.44 m). Looking further into the end of the century, under a compounded influence of high-water discharge (with a total volume exceedance frequency of approximately 20%), storm surge (with an annual exceedance probability of 1% AEP along the Bassac river mouth), land subsidence (averaging around 1.12 m ($\sigma$ =0.59 m)), and a SLR of

1 m, the projected inundation of the Camau peninsula is expected to encompass 99% of the area, with an average inundation depth of 2.20 m ($\sigma$ =0.68 m).

These results show that the future potential risks associated with the rising inundation levels in this study area need to be carefully considered concerning the future sustainability of the region and its ability to adapt to both current and forthcoming pressures due to human activity and climate change. This objective of these findings is to provide insight for strategic planners

as they contemplate various avenues for spatial development within the Ca Mau peninsula. They represent a crucial foundation for shaping policies and devising investment strategies related to infrastructure systems, including flood defences, tidal defence dykes, and irrigation and especially for safe transportation. Furthermore, the projected rise in inundation levels in this region, largely attributed to SLR, poses a significant challenge to the preservation of one of the pristine mangrove forest ecosystems in the VMD. This ecosystem is already under threat from ongoing infrastructure developments in the area. Therefore, it is

imperative to implement measurement, planning, and adaptation strategies to address this issue effectively in the future.


**6 Appendix A**

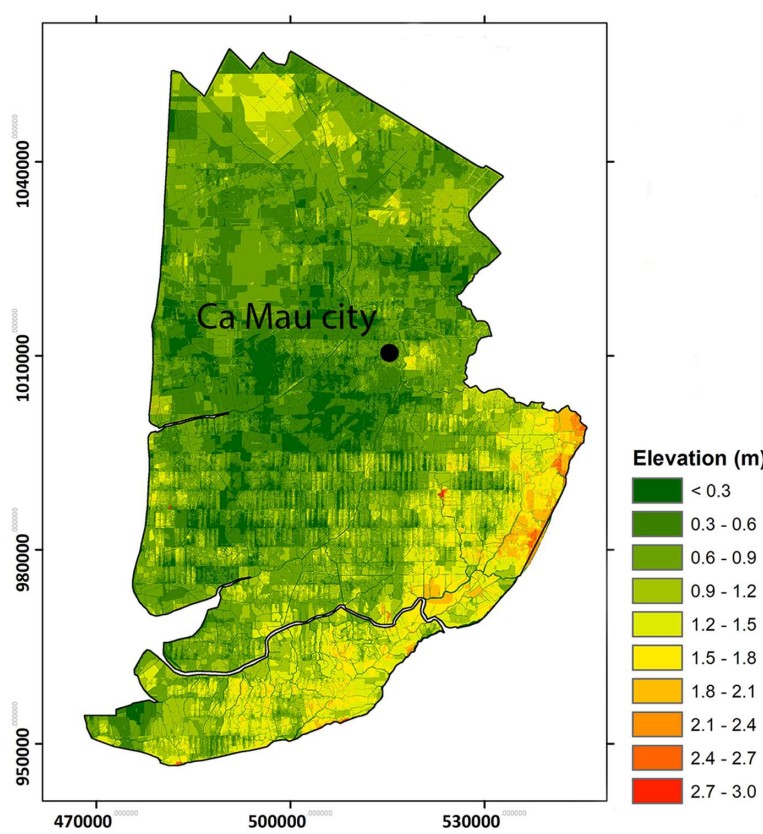

**Figure A1. DEM illustrates the flat terrain of the Ca Mau peninsula acquired from the Ministry of Natural Resources and**
**Environment (MONRE) in 2008**



*Author contribution.* **Hung Nguyen Nghia**: Conceptualization, Methodology, Resources, Visualization, Modelling work, Writing - original draft, Writing - review & editing. **Quan Quan Le**: Conceptualization, Methodology, Modelling work, Visualization, Writing - review & editing. **Dung Viet Nguyen**: Writing - review & editing. **Hong Tan Cao:** Writing - review & editing. **Toan Quang To:** review & editing. **Do Dac Hai:** Methodology, Writing - review & editing. **Melissa Wood**: Conceptualization, Methodology, Writing - review & editing. **Ivan D. Haigh**: Conceptualization, Methodology, Writing - review & editing

*Competing interests.* The authors declare that they have no conflict of interest

*Acknowledgments.* This study was co-funded by the UK National Environment Research Council (NERC) and the Viet Nam National Foundation for Science and Technology Development (NAFOSTED) under the grant number: NE/S003150/1. The authors thanks to Dr. Philip S.J. Minderhoud for his help in providing the land subsidence scenarios in the Mekong Delta.



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
