# Peer review of "Assessment of coastal inundation triggered by multiple drivers in Ca Mau Peninsula, Vietnam"

_Natural Hazards and Earth System Sciences, 2024_

## Author Comment (AC2)

**Review Rebuttal for manuscript entitled "Assessment of coastal inundation triggered by multiple drivers in Ca Mau Peninsula, Vietnam"**

We sincerely thank you for the thorough editorial work and for providing three comprehensive reviews. We greatly appreciate the opportunity to revise and refine the manuscript in response to the reviewers' insightful feedback. Below, we provide a point-by-point response to the reviewers' comments (in blue text), along with our rebuttals (in black text) or descriptions of the changes made to the manuscript (in black italicized text) in response to their observations.

**RC #1 (Remarks to the Author):**

The authors present a well-designed study modeling various flooding scenarios in the Ca Mau Peninsula. They have identified a research gap dealing with regional flooding dynamics, including compound flooding, and through the simulation plan, aim to contribute to filling this gap. Simulations were defined by discharge, storm surge occurrence, lowering of the river bed, land subsidence, and sea level rise, considering SSP5-8.5 scenarios. They conclude that land subsidence, rising sea levels, and their combined effects are the principal drivers behind increased inundation events. Overall I think the manuscript is of high quality with impactful results, my feedback deals mainly with the discussion of the modeling approach and the discussion section in general. I support the suggestions outlined in the community comment (CC1) and there is no need to add repeat suggestions. My review is structured as major points and line-by-line suggestions/feedback.

We deeply appreciate the reviewer's positive remarks on the strengths of our paper.

**Major points:**

 1) The limitations of 1D hydrodynamic modeling and not considering sediment transport should be discussed or further justified.

We appreciate the reviewer's valuable comment regarding the limitations of using a 1D hydrodynamic model and the exclusion of sediment transport in our study. However, we would like to provide further clarification and justification for our approach.

The 1D model we employed has been specifically designed to capture the complex interactions between the river and floodplain systems. This includes detailed representations of the main river, side channels, and floodplains. In particular, the model incorporates extensive data on secondary channels within the Vietnam Mekong Delta (VMD), which have been updated and integrated into the model. For instance, the model represents approximately 4,235 river branches and 564 floodplain compartments within VMD, as outlined by Dung, (2011). This configuration enables the model to effectively simulate dynamic water exchanges between the mainstream river and the floodplain system, including inundation patterns within the floodplain, as demonstrated in studies such as Manh et al., (2014) and Dung, (2011). Given this setup, we believe the model is well-suited to accurately capture the hydraulic regime as well as inundation dynamics in the Vietnam Mekong Delta (VMD),

particularly in the study area of the Ca Mau Peninsula. While the exclusion of sediment transport is acknowledged, this study partially addresses the limitation by incorporating predicted bathymetric changes in the main Mekong River up to 2038, considering sediment deficits and sand mining. These adjustments aim to capture certain aspects of sediment dynamics and their potential influence on the delta. However, it is important to note that the primary focus of this study is on hydrodynamics. The model is specifically configured to isolate and evaluate the multifaceted factors contributing to inundation in the Ca Mau Peninsula. This approach provides a clearer understanding of the scale and interplay of various drivers of flood hazards in the region.

To improve clarity and address this concern, we have included the following details in line 140-145:

*The model utilizes the WGS84 coordinate system, with elevations referenced to the Hon Dau Mean Sea Level (MSL). For the Mekong River's main channels and the Tonle Sap River, topographic data were compiled and updated through various projects of varying accuracy (Dung et al., 2011). The representation of floodplains differs between regions due to their unique characteristics. In the Cambodian Mekong Delta (CMD), where floodplains lack significant channelization and dike infrastructure, they are represented as channels with broad cross-sections (Dung et al., 2011). Conversely, in the Vietnamese Mekong Delta, the floodplains are highly compartmentalized into numerous flood cells to safeguard agricultural activities. These cells are depicted as wide cross-section channels enclosed by dikes (Dung et al., 2011).*

We have added a limitation within the discussion section, highlighting the study's constraints related to sediment transport. Specifically, line 478 now reads:

*In this study, flood risk in the Ca Mau Peninsula was assessed using 1D modelling, taking into account both individual and combined factors. Although the results provide useful insights, it is essential to recognize the inherent uncertainties associated with the model scenarios and methodologies used.*

*One limitation arises from the use of a one-dimensional model, which calculates water levels in the floodplain through spatial interpolation from the water levels in surrounding channels. While this approach may be less accurate compared to a 2D model, it still provides a reasonable estimate when combined with 2D modelling. Specifically, the 1D model utilized in this study incorporates extensive data on secondary channels within the Vietnam Mekong Delta (VMD), particularly within the Ca Mau Peninsula. The secondary channel data resolution ranges from 1 km to 3 km per channel in the study area, ensuring the model accurately represents the region's complex hydrodynamics. Moreover, the model includes approximately 4,235 river branches and 564 floodplain compartments across the VMD, as detailed by Dung et al., (2011). This configuration allows the model to effectively simulate dynamic water exchanges between the mainstream river and the floodplain system, including water level patterns within the floodplain, thus improving the representation of local hydrological conditions despite the limitations of the 1D approach. In addition, the model does not account for long-term sediment transport, which could influence changes in the floodplain area, with sedimentation potentially counteracting land subsidence (Hung et al., 2014; Zoccarato et al., 2018). While this limitation is acknowledged, the study partially addresses it by incorporating predicted bathymetric*

*changes in the main Mekong River up to 2038, considering factors such as sediment deficits and sand mining activities (Vasilopoulos et al., 2021). These adjustments are intended to capture certain aspects of sediment dynamics and their potential impact on the delta.*

*\*Dung, N. V., Merz, B., Bárdossy, A., Thang, T. D., & Apel, H. (2011). Multi-objective automatic calibration of hydrodynamic models utilizing inundation maps and gauge data. Hydrology and Earth System Sciences, 15, 1339–1354. https://doi.org/10.5194/hess-15-1339-2011*

*Manh, N. V., Dung, N. V., Hung, N. N., Merz, B., & Apel, H. (2014). Large-scale suspended sediment transport and sediment deposition in the Mekong Delta. Hydrology and Earth System Sciences, 18(8), 3033–3053. https://doi.org/10.5194/hess-18-3033-2014*

*Hung, N. N., Delgado, J. M., Güntner, A., Merz, B., Bárdossy, A., & Apel, H. (2014). Sedimentation in the fl oodplains of the Mekong Delta , Vietnam . Part I : suspended sediment dynamics. 3144, 3132–3144. https://doi.org/10.1002/hyp.9856*

*Zoccarato, C., Minderhoud, P. S. J., & Teatini, P. (2018). The role of sedimentation and natural compaction in a prograding delta: insights from the mega Mekong delta, Vietnam. Scientific Reports, 8(1), 1–12. https://doi.org/10.1038/s41598-018-29734-7*

*Vasilopoulos, G., Quan, Q. L., Parsons, D. R., Darby, S. E., Tri, V. P. D., Hung, N. N., Haigh, I. D., Voepel, H. E., Nicholas, A. P., & Aalto, R. (2021). Establishing sustainable sediment budgets is critical for climate-resilient mega-deltas. Environmental Research Letters, 16. https://doi.org/10.1088/1748-9326/ac06fc*

2) What were the roughness values used (lines 182)? Where were they different? And how do they compare to hydraulic manuals in terms of used values vs. guideline values (e.g. Chow, 1959)? If they are different from guideline values this should be explained.

We sincerely thank the reviewer for their insightful comment. The roughness values used in this model fall within ranges that are consistent with those reported in standard hydraulic textbooks and widely referenced publications (e.g. Chow et al., 1988; Werner et al., 2005). For further details on the selection and application of these values. See more detail in Dung et al.,2011, Manh et al., (2014), where these parameters are discussed extensively.

To enhance clarity and address this point, we have added the following information to line 182:

*The initial roughness values were based on Manh et al. (2014), who categorized the model domain into distinct zones, assigning specific Manning's n values to each zone to represent their unique hydraulic characteristics. These values were systematically refined to achieve the best agreement between the model's predicted water levels and discharges and the observed data. Specifically, the calibration process utilized monitoring stations located along the main Mekong channel in Vietnam (Fig. 2b, Table 1). Model outputs were compared with observed water levels and discharges, and any discrepancies prompted adjustments to the zone-specific roughness values. The model was iteratively rerun until optimal results were achieved. The final calibrated Manning's n values are detailed in Table 3.*

**Table 3.** The Manning roughness coefficient ($n$ ) is categorized based on different zones using in the modelling.  The Manning roughness in the channel changing for different cross-sections within the same zone

| Zone | Description | Manning's coefficient (n) |
| --- | --- | --- |
| 1 | Mekong River: Kratie to Phnom Penh | 0.032 |
| 2 | Mekong River: Phnom Penh to Tan Chau | 0.032 to 0.027 |
| 3 | Mekong River: Tan Chau to My Thuan | 0.027 to 0.025 |

| 4 | Mekong River: My Thuan to River months | 0.025 to 0.016 |
| 5 | Bassac River: Phnom Penh to Chau Doc | 0.032 |
| 6 | Bassac River: Chau Doc to Can Tho | 0.032 to 0.025 |
| 7 | Bassac River: Can Tho to River months | 0.025 to 0.017 |
| 8 | Side channels | 0.033 |
| 9 | Floodplain | 0.033 |

*Chow, V. Te, Maidment, D. R., & Mays, L. W. (1988). Applied Hydrology (pp. 1–294). McGraw-Hill, New York.

Werner, M. G. F., Hunter, N. M., & Bates, P. D. (2005). Identifiability of distributed floodplain roughness values in flood extent estimation. Journal of Hydrology, 314(1–4), 139–157. https://doi.org/10.1016/j.jhydrol.2005.03.012

Dung, N. V., Merz, B., Bárdossy, A., Thang, T. D., & Apel, H. (2011). Multi-objective automatic calibration of hydrodynamic models utilizing inundation maps and gauge data. Hydrology and Earth System Sciences, 15, 1339–1354. https://doi.org/10.5194/hess-15-1339-2011

Manh, V. N., Dung.V.N, Hung, N. ., Merz, B., & Apel, H. (2014). Large-scale quantification of suspended sediment transport and deposition in the Mekong Delta. Hydrology and Earth System Sciences Discussions, 18, 3033–3053. https://doi.org/10.5194/hess-18-3033-2014

3) Line 454-458: Potential mitigation measures, if discussed, require a much more thorough assessment with references. I think expanding on this section and connecting it to the natural (nature-based) storm surge buffering effects of mangrove forests would enhance this section while connecting to the importance of the ecological system. How would one establish a freshwater ecological zone within saline environments (lines 449-450)? Additionally, model limitations should be addressed in greater detail in the discussion section.

De Dominicis, M., Wolf, J., van Hespen, R. et al. Mangrove forests can be an effective coastal defence in the Pearl River Delta, China. Commun Earth Environ 4, 13 (2023). https://doi.org/10.1038/s43247-022-00672-7

We appreciate the reviewer's thoughtful and constructive suggestions. We concur that the discussion on potential mitigation measures could benefit from further expansion. Accordingly, we have revised and extended the discussion section to offer a more comprehensive assessment of these measures, while also incorporating the limitations. Section 4 now reads:

*4 Discussion*

*The aforementioned findings underscore the potential anthropogenic impact on the evolution of the inundation regime across the Ca Mau peninsula. These results suggest that the influences of riverbed lowering in the primary Mekong channel, and the occurrence of storm surges along the mainstream Mekong River are unlikely to significantly affect the inundation regime in this region, while high-water discharge did cause the inundation in this study area to rise only by approximately 0.1-0.3 m. The minimal impacts can be attributed to the prominence of high-water discharge, riverbed lowering, and storm surges primarily occurring in the upstream region and in the main Mekong and Bassac channels. Since the study area is situated farther from these channels and closer to the sea on both sides, these impacts tend to diminish. However, the results emphasizes that land subsidence and tidal influence combined with SLR will be the primary factors driving the escalation of the inundation regime in the Ca Mau peninsula in the future, both in extent and magnitude. In the projected future by the mid-century,*

under the S6 scenarios, a combination of high-water discharge, storm surge, land subsidence, and sea level rise are projected to inundate 96% of the Ca Mau Peninsula, with an average inundation depth of 0.91 m (σ = 0.44 m). Looking towards the end of the century under the S7 scenarios, the compounded effects of high-water discharge, storm surge, land subsidence, and a sea level rise are expected to inundate 99% of the Ca Mau Peninsula, with an average inundation depth of 2.20 m (σ = 0.68 m). However, it should be noted that these projections are based on the SSP5-8.5 scenarios, which represent the very worst case with low confidence. The ramifications of this increasing trend in inundation are now deliberated in relation to the region's future sustainability and its capacity to adapt to present and future transformations.

The increasing inundation resulting from land subsidence and SLR poses a significant threat of saltwater intrusion into inland regions, which will have severe consequences for agriculture, aquaculture, and infrastructure in the Ca Mau Peninsula. This area is not only economically important but also home to one of the most diverse natural ecosystems in the Mekong Delta, with over 79,100 hectares of mangrove forests that play a vital role in maintaining the region's ecological balance. These mangrove forests are essential for protecting coastal zones, regulating freshwater flow, and supporting local biodiversity. However, recent years have seen significant anthropogenic pressure on this ecosystem, particularly from the development of transportation infrastructure, irrigation systems, and water control structures. These developments have led to substantial degradation of the mangrove forests (Son et al., 2015; Hauser et al., 2020; Thuy et al., 2023), further compounded by the potential rise in inundation levels driven by land subsidence and SLR. This combination of factors threatens to accelerate the loss of these critical ecosystems, undermining their ability to provide natural coastal defence, support biodiversity, and mitigate the effects of climate change (Barbier et al., 2008; Temmerman et al., 2013; Jones et al. 2020; Sunkur et al., 2023).

Although land subsidence and SLR contribute to elevation loss and increased inundation, they may also create conditions favourable for sediment deposition in flooded areas. This sediment trapping mechanism has the potential to partially offset elevation loss over time, thereby influencing future inundation depths. However, the effectiveness of this natural compensatory process is significantly constrained by the sharp reduction in sediment flow from upstream rivers, largely due to damming (Kondolf et al., 2014). These dams have resulted in a projected 57% decrease in suspended sediment flux, from 99 Mt yr$^{-1}$ (1980-2009) to 43 Mt yr$^{-1}$ by 2020-2029 (Bussi et al., 2021). Sediment transport is crucial for maintaining coastal stability, replenishing eroded areas, and preserving elevation. The decline in sediment supply, driven by upstream dams and other human activities, has worsened the region's vulnerability, accelerating coastal erosion and shoreline retreat (Anthony et al., 2015; Tu et al., 2019).

In addition, the study also reveals concerns regarding the region's infrastructure, particularly road construction. According to the national North-South Expressway's design specifications for the period 2021-2025 (No: 912/QĐ-BGTVT), the road crest is set 2.2 m above mean sea level in the Ca Mau area (Fig. 2). However, land subsidence projections of approximately 0.77 m by mid-century and 2.34 m by

*the end of the century could lower the road crest to 1.43 m and -0.14 m, respectively. These reductions may not ensure adequate protection, especially considering predicted maximum water levels of approximately 1.59 m by 2050 and 2.20 m by 2100. This assessment is based on Vietnam's road construction standards (TCVN 4054: 2005). Furthermore, the development of infrastructure contributes additional pressure to the deltaic land surface, exacerbating land subsidence in these areas.*

*To mitigate the impacts of flooding caused by SLR and land subsidence in the Ca Mau Peninsula, it is crucial to implement a combination of sustainable strategies that address both the natural and anthropogenic drivers of vulnerability. A primary focus must be on reducing groundwater extraction, the leading cause of land subsidence in the region. This is critical not only for mitigating land subsidence but also for enhancing the area's resilience to the escalating impacts of climate change (Minderhoud et al., 2020). Additionally, increasing sediment flow from upstream rivers is essential for both replenishing eroded coastal areas and counteracting the effects of land subsidence. By improving sediment management in river systems, such as restoring natural sediment transport pathways and removing barriers like dams, the flow of sediment can be enhanced. Moreover, implementing water-saving technologies, such as efficient irrigation systems and the upgrading of water channels from the Bassac River, can significantly reduce the demand for groundwater in agriculture and domestic use. Moreover, developing alternative water sources, such as, rainwater harvesting, desalination, and surface water reservoirs, which can reduce dependency on groundwater, ensuring reliable water supplies during periods of drought or high demand.*

*In parallel, restoring and protecting the natural balance between saltwater and freshwater systems is essential. The creation or rehabilitation of tidal marshes and mangrove forests along the coastline can serve as natural barriers, filtering salinity, protecting against erosion and storm surges, and providing critical habitat for biodiversity while sustaining local livelihoods (Barbier et al., 2008;Temmerman et al., 2013; Jones et al., 2020; Sunkur et al., 2023; Dominicis et al., 2023). In addition, the integration of engineered infrastructure, including tidal barriers, levees, and dikes, can help prevent saltwater intrusion and manage flood risks, particularly in areas most vulnerable to the effects of SLR.*

*Community engagement is equally important in ensuring the long-term success of these strategies. Raising awareness about the negative impacts of excessive groundwater extraction and the benefits of sustainable water use can foster local participation in conservation efforts. Educational campaigns and training programs can also empower local stakeholders to adopt water-saving practices, contributing to the sustainable management of this vital resource. Finally, investing in robust monitoring systems is essential for tracking groundwater levels, land subsidence rates, and water usage patterns. Accurate, real-time data can inform effective decision-making, enabling authorities to prioritize interventions and manage resources efficiently. By integrating these natural, infrastructural, and community-based strategies, the Ca Mau Peninsula can build resilience against flooding, safeguard its ecosystems, and ensure the sustainability of its agricultural and aquacultural industries, even in the face of rising seas and land subsidence.*

In this study, flood risk in the Ca Mau Peninsula was assessed using 1D modelling, taking into account both individual and combined factors. Although the results provide useful insights, it is essential to recognize the inherent uncertainties associated with the model scenarios and methodologies used.

One limitation arises from the use of a one-dimensional model, which calculates water levels in the floodplain through spatial interpolation from the water levels in surrounding channels. While this approach may be less accurate compared to a 2D model, it still provides a reasonable estimate when combined with 2D modelling. Specifically, the 1D model utilized in this study incorporates extensive data on secondary channels within the Vietnam Mekong Delta (VMD), particularly within the Ca Mau Peninsula. The secondary channel data resolution ranges from 1 km to 3 km per channel in the study area, ensuring the model accurately represents the region's complex hydrodynamics. Moreover, the model includes approximately 4,235 river branches and 564 floodplain compartments across the VMD, as detailed by Dung et al., (2011). This configuration allows the model to effectively simulate dynamic water exchanges between the mainstream river and the floodplain system, including water level patterns within the floodplain, thus improving the representation of local hydrological conditions despite the limitations of the 1D approach. In addition, the model does not account for long-term sediment transport, which could influence changes in the floodplain area, with sedimentation potentially counteracting land subsidence (Hung et al., 2014; Zoccarato et al., 2018). While this limitation is acknowledged, the study partially addresses it by incorporating predicted bathymetric changes in the main Mekong River up to 2038, considering factors such as sediment deficits and sand mining activities (Vasilopoulos et al., 2021). These adjustments are intended to capture certain aspects of sediment dynamics and their potential impact on the delta.

Although the accuracy of topographic DEMs in the VMD remains a topic of ongoing discussion (Minderhoud et al., 2019), this study utilized a Digital Elevation Model (DEM) for the Ca Mau Peninsula developed by the Ministry of Natural Resources and Environment (MONRE) in 2008 (Tran et al., 2016). This DEM, with a resolution of 5 m × 5 m, is based on the WGS84 coordinate system and the Hon Dau mean sea level datum, and was created using high-quality survey data, topographic maps, and photogrammetric information (Tran et al., 2016). While this methodology aligns with the TOPO DEM by Minderhoud et al., (2019), which is known for its accuracy at a coarser resolution (500 m × 500 m), the higher resolution of our DEM is better suited for localized analyses of the Ca Mau Peninsula, providing greater precision in capturing the region's topography and inundation dynamics.

The use of a 5x5 m resolution DEM in this study has provided a detailed representation of the topographical variations within the study area. However, this high level of detail also introduces certain artefacts that become apparent in Figure 6 and 7. These artefacts stem from the granularity of the DEM dataset, which captures subtle variations in elevation between adjacent cells. Such elevation differences are particularly pronounced in the mapping of inundation levels, where thresholds such as <0.1 m (unflooded; white colour, Fig. 6) and 0.1–0.4 m (flooded; green colour, Fig. 6) highlight the varying extents of inundated areas. While this granularity allows for a more precise delineation of inundation zones, it also introduces abrupt transitions in the inundation extent that may not accurately

reflect real-world conditions. These artefacts are especially noticeable in flat or low-lying areas, where small discrepancies in elevation can disproportionately affect the modelled inundation extent. Despite these limitations, the 5x5 m DEM provides valuable insights into the spatial distribution of inundation areas at a resolution suitable for regional-scale assessments. The artefacts observed in this study highlight the importance of balancing DEM resolution with the study's scale and objectives to ensure accurate and reliable outcomes.

The assumption that areas with a depth of less than 0.1 m are classified as unflooded is typically used to identify negligible inundation. However, in urban areas, even shallow water depths can become hazardous when associated with high flow velocities. Such conditions can lead to risks, such as infrastructure damage, heightened erosion, and potential threats to public safety.

Additionally, the use of a DEM from 2008 introduces a temporal discrepancy when combined with sea level rise (SLR) and land subsidence scenarios derived from 2018 data. Between 2008 and 2018, SLR in the coastal region was relatively small, with an estimated increase of less than 5 cm according to MONRE projections (Tran et al., 2016), which had minimal impact on the analysis. However, land subsidence during this period was more pronounced, with a maximum of 15 cm in certain areas (Minderhoud et al., 2017). Although these changes could introduce slight inaccuracies in the DEM's topography, the variations in relative elevation from 2008 to 2018 remain negligible within the scope of this study. Specifically, sea level rise scenarios range from 0.5 m to 1 m, and land subsidence scenarios range from 0.38 m ($\sigma$ = 0.21 m) to 1.12 m ($\sigma$ = 0.59 m), reflecting regional variability. Thus, the relative elevation changes from 2008 to 2018 are considered minimal within the scope of our analysis.

Regarding land subsidence scenarios, this study focuses solely on the impact of groundwater extraction, excluding other factors contributing to land subsidence, such as natural subsidence, tectonic movements, or other human activities (Minderhoud et al., 2017; Zoccarato et al., 2018; Karlsrud et al., 2020). This emphasis is based on the fact that groundwater extraction is the primary driver of land subsidence in the Ca Mau Peninsula (Minderhoud et al., 2017;  Karlsrud et al., 2020).

However, it is important to acknowledge that reduced sediment input from upstream could exacerbate natural subsidence processes, particularly through natural compaction, as discussed by Zoccarato et al., (2018). Therefore, while this study does not account for these additional subsidence factors, the broader context of sediment deficits should be considered as a contributing factor to the overall land subsidence in the region.

Finally, the inundation scenarios in this study are based on long-term projections that consider the cumulative impacts of land subsidence and sea level rise up to 2100. While these projections are essential for evaluating future risks and guiding long-term planning, they may not fully account for short-term fluctuations or temporary environmental changes, potentially leading to an overestimation of short-term impacts. Although the focus on long-term trends offers valuable insights into future scenarios, further research that includes short-term variability is needed to provide a more

*comprehensive understanding of the region's vulnerability to flooding and land subsidence in the near term.*

*Anthony, E. J., Brunier, G., Besset, M., Goichot, M., Dussouillez, P., & Nguyen, V. L. (2015). Linking rapid erosion of the Mekong River delta to human activities. Scientific Reports, 5. https://doi.org/10.1038/srep14745*

*Barbier, E. B., Koch, E. W., Silliman, B. R., Hacker, S. D., Wolanski, E., Primavera, J., Granek, E. F., Polasky, S., Aswani, S., Cramer, L. A., Stoms, D. M., Kennedy, C. J., Bael, D., Kappel, C. V., Perillo, G. M. E., & Reed, D. J. (2008). Coastal ecosystem-based management with nonlinear ecological functions and values. Science, 319(5861), 321–323. https://doi.org/10.1126/science.1150349*

*Bussi, G., Darby, S. E., Whitehead, P. G., Jin, L., Dadson, S. J., Voepel, H. E., Vasilopoulos, G., Hackney, C. R., Hutton, C., Berchoux, T., Parsons, D. R., & Nicholas, A. (2021). Impact of dams and climate change on suspended sediment flux to the Mekong delta. Science of the Total Environment, 755. https://doi.org/10.1016/j.scitotenv.2020.142468*

*De Dominicis, M., Wolf, J., van Hespen, R., Zheng, P., & Hu, Z. (2023). Mangrove forests can be an effective coastal defence in the Pearl River Delta, China. Communications Earth and Environment, 4(1). https://doi.org/10.1038/s43247-022-00672-7*

*Dung, N. V., Merz, B., Bárdossy, A., Thang, T. D., & Apel, H. (2011). Multi-objective automatic calibration of hydrodynamic models utilizing inundation maps and gauge data. Hydrology and Earth System Sciences, 15, 1339–1354. https://doi.org/10.5194/hess-15-1339-2011*

*Hauser, L. T., Hauser, L. T., Binh, N. A., Hoa, P. V., Quan, N. H., & Timmermans, J. (2020). Gap-Free Monitoring of Annual Mangrove Forest Dynamics in Ca Mau Province , Vietnamese Mekong Delta , Using the Landsat-7-8 Archives and Post-Classification Temporal Optimization. MDPI. https://doi.org/doi:10.3390/rs12223729*

*Hung, N. N., Delgado, J. M., Güntner, A., Merz, B., Bárdossy, A., & Apel, H. (2014). Sedimentation in the fl oodplains of the Mekong Delta , Vietnam . Part I : suspended sediment dynamics. 3144, 3132–3144. https://doi.org/10.1002/hyp.9856*

*Jones, H. P., Nickel, B., Srebotnjak, T., Turner, W., Gonzalez-Roglich, M., Zavaleta, E., & Hole, D. G. (2020). Global hotspots for coastal ecosystem-based adaptation. PLoS ONE, 15(5), 1–17. https://doi.org/10.1371/journal.pone.0233005*

*Karlsrud, K., Tunbridge, L., Khanh, N. Q., & Dinh, N. Q. (2020). Preliminary results of land subsidence monitoring in the Ca Mau Province. Proceedings of the International Association of Hydrological Sciences, 382, 111–115. https://doi.org/10.5194/piahs-382-111-2020*

*Kondolf, G. M., Rubin, Z. K., & Minear, J. T. (2014). Dams on the Mekong: Cumulative sediment starvation. Water Resources Research, 50, 5158–5169. https://doi.org/10.1002/2013WR014651*

*Minderhoud, P. S. J., Coumou, L., Erkens, G., Middelkoop, H., & Stouthamer, E. (2019). Mekong delta much lower than previously assumed in sea-level rise impact assessments. Nature Communications, 1–13. https://doi.org/10.1038/s41467-019-11602-1*

*Minderhoud, P. S. J., Erkens, G., Pham, V. H., Bui, V. T., Erban, L., Kooi, H., & Stouthamer, E. (2017). Impacts of 25 years of groundwater extraction on subsidence in the Mekong delta, Vietnam. Environmental Research Letters, 12. https://doi.org/10.1088/1748-9326/aa7146*

*Minderhoud, P. S. J., Middelkoop, H., Erkens, G., & Stouthamer, E. (2020). Groundwater extraction may drown mega-delta : projections of extraction-induced subsidence and elevation of the Mekong delta for the 21st century. Environmental Research Letters. https://doi.org/https://doi.org/10.1088/2515-7620/ab5e21*

*Son, N., Chen, C., Chang, N., Chen, C., Chang, L., & Thanh, B. (2015). Mangrove Mapping and Change Detection in Ca Mau Peninsula , Vietnam , Using Landsat Data and Object-Based Image Analysis. IEEE Journal of Selected Topics in Applied Earth Observations and Remote Sensing, 8(2), 503–510. https://doi.org/10.1109/JSTARS.2014.2360691*

*Sunkur, R., Kantamaneni, K., Bokhoree, C., & Ravan, S. (2023). Mangroves' role in supporting ecosystem-based techniques to reduce disaster risk and adapt to climate change: A review. Journal of Sea Research, 196(October), 102449. https://doi.org/10.1016/j.seares.2023.102449*

*Temmerman, S., Meire, P., Bouma, T. J., Herman, P. M. J., Ysebaert, T., & De Vriend, H. J. (2013). Ecosystem-based coastal defence in the face of global change. Nature, 504(7478), 79–83. https://doi.org/10.1038/nature12859*

*Thuy, L., Nguyen, M., Thi, H., Choi, E., & Sun, P. (2023). Estuarine , Coastal and Shelf Science Distribution of mangroves with different aerial root morphologies at accretion and erosion sites in Ca Mau Province , Vietnam. Estuarine, Coastal and Shelf Science, 287(March), 108324. https://doi.org/10.1016/j.ecss.2023.108324*

*Tran, T., Nguyen, V., Huynh, T., Mai, V., Nguyen, X., & HP, D. (2016). Kịch bản biến đổi khí hậu và nước biển dâng cho Việt Nam (Climate change and sea level rise scenarios for Vietnam). Nhà xuất bản Tài Nguyên Môi Trường và Bản Đồ Việt Nam, Bộ Tài Nguyên và Môi Trường, Hà Nội, p 188 [Publishing House of Natural Resources, Enviroment and Cartography, Ministry of Natural Resources and Environment, Ha Noi, p 188]. http://www.imh.ac.vn/files/doc/KichbanBDKH/KBBDKH_2016.pdf*

*Tu, L. X., Thanh, V. Q., Reyns, J., Van, S. P., Anh, D. T., Dang, T. D., & Roelvink, D. (2019). Sediment transport and morphodynamical modeling on the estuaries and coastal zone of the Vietnamese Mekong Delta. Continental Shelf Research, 64–76. https://doi.org/10.1016/j.csr.2019.07.015*

*Vasilopoulos, G., Quan, Q. L., Parsons, D. R., Darby, S. E., Tri, V. P. D., Hung, N. N., Haigh, I. D., Voepel, H. E., Nicholas, A. P., & Aalto, R. (2021). Establishing sustainable sediment budgets is critical for climate-resilient mega-deltas. Environmental Research Letters, 16. https://doi.org/10.1088/1748-9326/ac06fc*

*Zoccarato, C., Minderhoud, P. S. J., & Teatini, P. (2018). The role of sedimentation and natural compaction in a prograding delta: insights from the mega Mekong delta, Vietnam. Scientific Reports, 8(1), 1–12. https://doi.org/10.1038/s41598-018-29734-7*

**4) Line by Line Suggestions:**

Line 16: I would remove historically since been implies past tense.

We have made the edit

Line 32: Remove comma after area

We have made the edit

Line 47: Ward et al., 2018 after Wahl et al., 2018

We have made the edit

Line 55: Suggest amplify instead of worsen

We have made the edit

Lines 57-58: I would add and before "to facilitate"

We have made the edit

Line 66: References in chronological order

We have made the edit

Line 66: Suggest: …, which is a major concern… (keep as 1 sentence)

We have made the edit

Line 69: Suggest: I would remove penetration since intrusion inland describes the same thing or you could say the extent of the inland tidal intrusion

We have made the edit

Line 73: the IPCC report

We have made the edit

Line 86: reference prior research

We have made the edit

Line 116: Currently

We have made the edit

Line 304: DEM of the region

We have made the edit

Line 305: ArcGIS model builder?

We have made the edit

Line 405: why not put the units with the number in figure 5 (and 6)?

We have made the edit

Line 477: delete with after with

Many thanks for your feedback; I have incorporated these changes accordingly.

**RC2 (Remarks to the Author):**

This study aims to investigate multiple drivers of the compound flood risk in Ca Mau Peninsula, Vietnam using a 1D hydrodynamic model under different simulation scenarios. Results indicated that the primary drivers for the escalation of inundation events in the study area are land subsidence, rising sea levels, and their combined effects. Overall, the study is comprehensive, and the findings are meaningful. The community member and the other reviewer have provided constructive comments, upon which I have several additional concerns and suggestions regarding the methodology and results of this study as follows.

We sincerely appreciate the reviewer's positive feedback and for highlighting the importance of our work.

**Methodology**

1) 1D hydrodynamic models have some limitations compared to 2D models especially when we are interested in the flood inundation extents for relatively flat floodplains and urban areas. How did you justify the validity of applying a 1D model to the study area?

We thank the reviewer for their valuable feedback, however, as noted in our response to Point 1, RC1, we acknowledge the limitations of using a 1D hydrodynamic model, particularly when compared to 2D models, which can provide more detailed representations of flood inundation extents in floodplains areas. However, we believe that the 1D model used in this study is highly suitable due to several key factors as we presented bellow:

Firstly, the 1D model we employed here is specifically designed to capture the complex interactions between the river system and floodplains throughout the entire Lower Mekong Basin. This includes not only the primary river channels but also secondary channels and the floodplains, which are integral components of the hydraulic regime. The model accurately represents the floodplain dynamics, including the interactions between the main river and floodplain areas. For more detail, two-dimensional profiles derived from ground survey data are used to represent the expansive natural floodplains of the Mekong River in the upper delta, located within Cambodia and around Tonle Sap. In contrast, shorter cross-sections, incorporating attached storage cells, are employed in the lower delta to model the canalized channels bordered by dykes and adjacent to traditional terraced paddies found in Vietnam (Dung et al.,2011). These features make the model effective for simulating inundation dynamics, even in areas with relatively flat topography, such as the Ca Mau Peninsula.

Moreover, the initial version of this model was validated for flood inundation extents in the Mekong Delta, particularly for the year 2008, as detailed in Dung et al. (2011). The validation of the model against observed data further supports its ability to accurately predict flood inundation in the study area. Based on these previous validations and its capacity to represent the essential hydrodynamic processes in the region, we are confident that the 1D model is well-suited for this study.

While 2D models might offer additional precision in highly detailed urban settings or particularly flat floodplains, the 1D model provides an effective and practical approach for understanding the broader,

region-wide dynamics of flooding in the Mekong Delta, especially in the context of our focus on the Ca Mau Peninsula. Therefore, we believe the application of the 1D model is both justified and appropriate for the objectives of this study.

We have expanded the limitations section within the discussion, highlighting the study's constraints regarding sediment transport. Specifically, line 478 in the Discussion section now states:

*In this study, flood risk in the Ca Mau Peninsula was assessed using 1D modelling, taking into account both individual and combined factors. Although the results provide useful insights, it is essential to recognize the inherent uncertainties associated with the model scenarios and methodologies used.*

*One limitation arises from the use of a one-dimensional model, which calculates water levels in the floodplain through spatial interpolation from the water levels in surrounding channels. While this approach may be less accurate compared to a 2D model, it still provides a reasonable estimate when combined with 2D modelling. Specifically, the 1D model utilized in this study incorporates extensive data on secondary channels within the Vietnam Mekong Delta (VMD), particularly within the Ca Mau Peninsula. The secondary channel data resolution ranges from 1 km to 3 km per channel in the study area, ensuring the model accurately represents the region's complex hydrodynamics. Moreover, the model includes approximately 4,235 river branches and 564 floodplain compartments across the VMD, as detailed by Dung et al., (2011). This configuration allows the model to effectively simulate dynamic water exchanges between the mainstream river and the floodplain system, including water level patterns within the floodplain, thus improving the representation of local hydrological conditions despite the limitations of the 1D approach. In addition, the model does not account for long-term sediment transport, which could influence changes in the floodplain area, with sedimentation potentially counteracting land subsidence (Hung et al., 2014; Zoccarato et al., 2018). While this limitation is acknowledged, the study partially addresses it by incorporating predicted bathymetric changes in the main Mekong River up to 2038, considering factors such as sediment deficits and sand mining activities (Vasilopoulos et al., 2021). These adjustments are intended to capture certain aspects of sediment dynamics and their potential impact on the delta.*

*Dung, N. V., Merz, B., Bárdossy, A., Thang, T. D., & Apel, H. (2011). Multi-objective automatic calibration of hydrodynamic models utilizing inundation maps and gauge data. Hydrology and Earth System Sciences, 15, 1339–1354. https://doi.org/10.5194/hess-15-1339-2011*

*Hung, N. N., Delgado, J. M., Güntner, A., Merz, B., Bárdossy, A., & Apel, H. (2014). Sedimentation in the fl oodplains of the Mekong Delta , Vietnam . Part I : suspended sediment dynamics. 3144, 3132–3144. https://doi.org/10.1002/hyp.9856*

*Vasilopoulos, G., Quan, Q. L., Parsons, D. R., Darby, S. E., Tri, V. P. D., Hung, N. N., Haigh, I. D., Voepel, H. E., Nicholas, A. P., & Aalto, R. (2021). Establishing sustainable sediment budgets is critical for climate-resilient mega-deltas. Environmental Research Letters, 16. https://doi.org/10.1088/1748-9326/ac06fc*

*Zoccarato, C., Minderhoud, P. S. J., & Teatini, P. (2018). The role of sedimentation and natural compaction in a prograding delta: insights from the mega Mekong delta, Vietnam. Scientific Reports, 8(1), 1–12. https://doi.org/10.1038/s41598-018-29734-7*

2) The definition of risks may be slightly different in literature, either including three primary factors, i.e., hazard, vulnerability, and exposure, or the probability of extreme events and the consequences due to their occurrence. However, this study mainly focused on the possible scenarios of flood

inundation depths and extents (or hazards) rather than the consequences due to the floods. I suggest authors keep the terms (e.g., risk and vulnerability) consistent with what is commonly used in relevant literature.

Many thanks for your helpful feedback. Based on your suggestions, we have made adjustments to ensure consistency with commonly used terminology in the literature.

Line 94 now read: *Recognizing the profound significance of this inquiry, particularly in relation to the heightened flood risks faced by local communities.*

Line 475 now read: *The Ca Mau peninsula is identified as one of the most promising economic hubs for agriculture and aquaculture within both the Mekong Delta region and Vietnam as a whole. Nonetheless, these areas also experience considerable flood risk.*

3) What is the time interval used in the model simulation? Hourly and Daily, or adjustable time steps based on the stability of the hydrodynamic model?

In our modelling, the simulations are run with a 5-minute time step. We will add this information on line 142:

*The model operates on a 5-minute time step, ensuring the stability of the hydrodynamic model and allowing for accurate and dynamic simulations of hydrodynamic processes.*

4) Previous studies have shown that the roughness parameter is a key factor in 1D flood modeling and roughness coefficients tend to change at different water depths. Even though the model used in this study has been fully calibrated, how did you make sure the calibrated parameters are still applicable under extreme flood events?

We sincerely thank the reviewer for their insightful comment. The roughness values used in this study fall within ranges that are consistent with those found in standard hydraulic textbooks and widely cited publications (e.g., Chow et al., 1988; Werner et al., 2005). For additional details on the selection and application of these values, we refer to Dung et al. (2011), where these parameters are discussed in depth.

We acknowledge that this is a challenging issue to address. We understand that the hydraulic roughness parameter is difficult to estimate (Lane, 2005; Ferguson, 2010). The roughness is influenced by numerous factors, including bed material, channel cross-section variation, vegetation, surface irregularities, obstructions, and channel sinuosity, among others (Gordon et al., 2004). This complexity makes it difficult to identify a single roughness value that accurately represents the surface roughness under all conditions.

In our study, we calibrated the roughness values based on observed data for typical flood conditions, acknowledging that roughness coefficients can vary with water depth. However, as with many 1D

hydrodynamic models, incorporating depth-dependent changes in roughness is challenging. The model treats the hydraulic roughness as a constant value, which simplifies the computation. While dynamic adjustment of roughness values based on water depth is a known factor in flood modeling, it was not feasible to incorporate such changes into our study, given the focus and constraints of the 1D model.

It is also important to note that the study area, located in a delta- coastal region, is subject to relatively smaller fluctuations in water levels compared to more upstream areas. In such coastal environments, water levels tend to rise gradually during the flood season, with less abrupt variation in response to changes in upstream water flux. This characteristic of the region implies that extreme flood events, while significant, do not exhibit the same dramatic shifts in water levels that might be expected in areas further upstream, where larger variations in discharge and water depth occur. Given this, the calibrated roughness values are considered sufficiently robust for modelling flood inundation dynamics in the study area, including during extreme flood events.

*Dung, N. V., Merz, B., Bárdossy, A., Thang, T. D., & Apel, H. (2011). Multi-objective automatic calibration of hydrodynamic models utilizing inundation maps and gauge data. Hydrology and Earth System Sciences, 15, 1339–1354. https://doi.org/10.5194/hess-15-1339-2011

Ferguson, R. (2010). Time to abandon the Manning equation? Earth Surface Processes and Landforms, 35(15), 1873–1876. https://doi.org/10.1002/esp.2091

Gordon, N. D., McMahon, T. A., Finlayson, B. L., Gippel, C. J., & Nathan, R. J. (2004). Stream Hydrology: An Introduction for Ecologists. Wiley. https://books.google.co.uk/books?id=_PJHw-hSKGgC

Lane, S. N. (2005). Roughness - Time for a re-evaluation? Earth Surface Processes and Landforms, 30, 251–253. https://doi.org/10.1002/esp.1208

Werner, M. G. F., Hunter, N. M., & Bates, P. D. (2005). Identifiability of distributed floodplain roughness values in flood extent estimation. Journal of Hydrology, 314(1–4), 139–157. https://doi.org/10.1016/j.jhydrol.2005.03.012

5) NSE, deviation, and $R^2$ were employed in the flood model calibration and validation processes. However, the weaknesses of these metrics should be noted, and it is suggested to apply the metrics to the flow periods of interest (high flows in the case) and present the values of metrics with a statistical distribution instead of a fixed number given the sampling uncertainty. The authors can refer to the article below for more information about the limitations of these evaluation metrics.

Reference: "Beyond a fixed number: Investigating uncertainty in popular evaluation metrics of ensemble flood modeling using bootstrapping analysis" (https://doi.org/10.1111/jfr3.12982)

We appreciate the reviewer's insightful feedback. While we acknowledge the limitations of the performance metrics (NSE, $R^2$), these were chosen based on their established use in hydrological modelling. $R^2$ indicates the proportion of variance in the observed data explained by the model, with values greater than 0.5 generally deemed acceptable (Moriasi et al., 2007). The Nash-Sutcliffe Efficiency (NSE) quantifies the degree of agreement between observed and predicted values, with an NSE of 1 indicating perfect model performance (Nash and Sutcliffe, 1970). While these metrics have strengths, such as providing a normalized measure of model accuracy, they are sensitive to extreme values and may not fully capture model performance under highly skewed conditions (Legates and McCabe, 1999; Huang and Merwade, 2024). For instance, high-flow events can introduce uncertainty in the NSE and $R^2$, especially in regions with skewed flow data (Clark et al., 2021; Duc and Sawada,

2023). However, as outlined in our response to point 4, this study employed ($NSE$, $R^2$) as model performance metrics due to their common usage in similar flood modelling contexts and their suitability for the relatively stable flow data observed in the study area. The hydraulic regime (Water discharge and water level) in delta- coastal region exhibit minimal skewness, making $NSE$ and $R^2$ appropriate choices for model calibration and validation.

However, we have incorporated the suggested information into line 185, now read:

*The Nash-Sutcliffe model efficiency coefficient (NSE) (Nash and Sutcliffe, 1970), as specified in Eq. (1), along with the coefficient of determination ($R^2$), as defined in Eq. (2), as defined in Eq. (2), which indicates the proportion of variance in both simulated and observed data explained by the model (Moriasi et al., 2007). was utilized to assess the accuracy of the comparison between the recently calibrated predictions and the observed data*

$$NSE = 1 - \frac{\sum_{t=1}^{T}(X_m^t - X_0^t)^2}{\sum_{t=1}^{T}(X_0^t - \overline{X_0})^2} \tag{1}$$

$$R^2 = \frac{\left[\sum_{t=1}^{T}(X_m^t - \overline{X_m})(X_0^t - \overline{X_0})\right]^2}{\sum_{t=1}^{T}(X_m^t - \overline{X_m})^2 \sum_{t=1}^{T}(X_0^t - \overline{X_0})^2} \tag{2}$$

*In this equation, $\overline{X_0}$ represents the mean of observed values and $X_0^t$ is the observed data at time t, $\overline{X_m}$ represents the mean of calibrated value, $X_m^t$ is the calibrated value at time t, Generally, NSE values below 0.5 indicate suboptimal calibration performance, while NSE values surpassing 0.5 suggest a satisfactory model performance. NSE values exceeding 0.65 indicate a well-performing calibration, and values surpassing 0.8 indicate a highly accurate calibration (Ritter and Muñoz-carpena, 2013), $R^2$ ranges from 0 to 1, with higher values indicating less error variance (Moriasi et al., 2007).*

And in caption of figure 4:

*Figure 4. Assessing the agreement between the predicted and observed hourly data for water level and water discharge at monitoring stations throughout the VMD during the calibration run in 2018 and validation run in 2016 along with the coefficient of determination ($R^2$).*

Nash, J. E., & Sutcliffe, J. V. (1970). River flow forecasting through conceptual models part I - A discussion of principles. Journal of Hydrology, 10, pp: 282-290.

Legates, D. R., & McCabe, G. J. (1999). Evaluating the use of goodness-of-fit Measures in hydrologic and hydroclimatic model validation. In Water Resources Research (Vol. 35, pp. 233–241).

Moriasi, D. N., Arnold, J. G., Liew, M. W. V., Bingner, R. L., Harmel, R. D., & Veith, T. L. (2007). Model evaluation guidelines for systematic quantification of accuracy in watershed simulations. American Society of Agricultural and Biological Engineers, 50, 885–900.

Ritter, A., & Muñoz-carpena, R. (2013). Performance evaluation of hydrological models : Statistical significance for reducing subjectivity in goodness-of-fit assessments. Journal of Hydrology, 480, 33–45. https://doi.org/10.1016/j.jhydrol.2012.12.004

Clark, M. P., Vogel, R. M., Lamontagne, J. R., Mizukami, N., Knoben, W. J. M., Tang, G., Gharari, S., Freer, J. E., Whitfield, P. H., Shook, K. R., & Papalexiou, S. M. (2021). The Abuse of Popular Performance Metrics in Hydrologic Modeling. Water Resources Research, 57, 1–16. https://doi.org/10.1029/2020WR029001

Duc, L., & Sawada, Y. (2023). A signal-processing-based interpretation of the Nash-Sutcliffe efficiency. Hydrology and Earth System Sciences, 27, 1827–1839. https://doi.org/10.5194/hess-27-1827-2023

Huang, T., & Merwade, V. (2024). Beyond a fixed number: Investigating uncertainty in popular evaluation metrics of ensemble flood modeling using bootstrapping analysis. Journal of Flood Risk Management, 1–20.

*https://doi.org/10.1111/jfr3.12982*

6) In Figure 4, what is the (linear) regression model the R$^2$ is measuring? Also, please note that R^2 may not capture the bias in the model prediction.

Many thanks to the reviewer for their valuable feedback. As response to point 5 above, we believe that two performance metrics:NSE, and $R^2$ is suitable to assess the hydraulic performance in delta-coastal region.

We have made change in caption of figure 4:

*Figure 4. Assessing the agreement between the predicted and observed hourly data for water level and water discharge at monitoring stations throughout the VMD during the calibration run in 2018 and validation run in 2016 along with the coefficient of determination ($R^2$).*

7) Line 304: "Areas with a depth < 0.1 m are classified as unflooded." What is the basis of this assumption? The water depth in Figure 1 (right) may be less than 0.1 m. Flood water with a depth of less than 0.1 m but with a high velocity can be dangerous in urban areas.

We appreciate the reviewer's comment and recognize the importance of addressing the assumption related to areas with a water depth of less than 0.1 m. In this study, we classified areas with a depth of <0.1 m as unflooded based on the practical consideration that such shallow water typically represents residual surface water or minor pooling, which does not pose a significant flood hazard under normal conditions. This threshold is consistent with standard practices in flood modelling (Balica et al., 2014), where a minimum depth is set to differentiate meaningful inundation from negligible water levels caused by model uncertainties, interpolation artifacts, or ground irregularities. However, we acknowledge that in urban areas, floodwater with a depth of <0.1 m can still present risks, especially when accompanied by high velocities, which can exacerbate hazard conditions. While our model primarily uses water depth to estimate flood extent, it does not explicitly account for floodplain velocity in the classification, due to the inherent limitations of the 1D modelling framework.

To provide a more comprehensive response to the reviewer's concern, we have incorporated additional information acknowledging this limitation into line 576 in discussion section, which now reads as follows.

*The assumption that areas with a depth of less than 0.1 m are classified as unflooded is typically used to identify negligible inundation. However, in urban areas, even shallow water depths can become hazardous when associated with high flow velocities. Such conditions can lead to risks, such as infrastructure damage, heightened erosion, and potential threats to public safety.*

*Balica, S., Dinh, Q., Popescu, I., Vo, T. Q., & Pham, D. Q. (2014). Flood impact in the Mekong Delta, Vietnam. Journal of Maps, 10, 257–268. https://doi.org/10.1080/17445647.2013.859636*

**Results**

8) Figure 4: Why are some data points of water discharge negative?

We appreciate the reviewer's comment. The negative discharge values observed in the delta region are a result of tidal oscillations and backwater effects. These phenomena can cause reverse flow, which leads to negative discharge values during certain tidal cycles.

9) Figure 5: The maps for S4_a and S4_b scenarios are missing?

We thank the reviewer for their comment. The S4_a and S4_b scenarios, which represent land subsidence, have been added to Figure 5.

10) Table 6: The accumulated increase in flooded areas for S1 at the level of 0.1-0.4 m is 43.0%? At least it is not true based on the results in Table 5. Please double-check the results presented in the tables, which will affect your conclusions.

We appreciate the reviewer's feedback. The values presented in Table 6 represent the accumulated increase in flooded areas for different flood drivers compared to the baseline scenario (S0).

For example, in area A, the inundation depth for S0 scenario is 0.5 m, while for S1 scenario is 0.6 m. Therefore, the increase in inundated area for S1 relative to S0 is 0.1 m, representing the cumulative change in flooded areas due to the different flood drivers compared to S0.

To improve clarity and address the reviewer's feedback, we have added the term "S0" to the caption of Table 6. It now reads:

*"Table 6. The percentage increase in area of flooding in each model scenario, compared to the baseline scenario (S0) at different levels in the Ca Mau Peninsula region"*

**Minor Issues**

11) It is suggested to add a north arrow and a scale bar to Figure 2, Figures 5-7, and Figure A1. What are the units of the numbers along the box in Figures 2(a) and A1?

We appreciate the reviewer's feedback and have made the necessary adjustments.

12) Line 273: What do the numbers in "B1.5" and "B2" stand for?

We thank the reviewer for their feedback. The numbers "B1.5" and "B2" refer to specific land subsidence scenarios developed to simulate future groundwater extraction pathways in the Mekong Delta of Vietnam from 2019 to 2100 under different mitigation conditions. These scenarios are secondary data adopted from the work of Minderhoud et al. (2020) through personal communication.

We have updated the manuscript to clarify these scenarios further, the line 273 now read:

***Delta Subsidence (S4_a, S4_b):*** *These scenarios evaluate the changes in inundation associated with land subsidence in the future. In this study, we only assess the impact of groundwater extraction and do not include other components of land subsidence and does not include other contributing factors such as natural subsidence, tectonic movements, or other human activities (Minderhoud et al., 2017;*

*Zoccarato et al., 2018); Karlsrud et al., 2020). This focus is due to the fact that land subsidence in the Ca Mau Peninsula is primarily driven by groundwater extraction (Minderhoud et al., 2017; Karlsrud et al., 2020).*

*The spatial map depicting future land subsidence within the VMD is derived from Minderhoud et al., (2020). This dataset includes land subsidence scenarios labeled B1.5 for the year 2050 ($B1.5_{2050}$) and B2 for the year 2100 ($B2_{2100}$). The B1.5 scenario simulates land subsidence under a moderate increase in groundwater extraction, serving as a pathway between the B1 scenario (moderate increase) and the B2 scenario (extreme increase). Both scenarios follow a non-mitigation pathway, where groundwater extraction continues to increase without any reduction measures, leading to greater land subsidence over time (Minderhoud et al., 2020). In this study, we use the land subsidence scenarios $B1.5_{2050}$ and $B2_{2100}$, which are referred to as the S4_a and S4_b scenarios, respectively. The S4_a scenario represents an average land subsidence of 0.38 m (σ = 0.21 m), while the S4_b scenario reflects 1.12 m (σ = 0.59 m) of subsidence in the Ca Mau Peninsula.*

*\*Karlsrud, K., Tunbridge, L., Khanh, N. Q., & Dinh, N. Q. (2020). Preliminary results of land subsidence monitoring in the Ca Mau Province. Proceedings of the International Association of Hydrological Sciences, 382, 111–115. https://doi.org/10.5194/piahs-382-111-2020*

*Minderhoud, P. S. J., Erkens, G., Pham, V. H., Bui, V. T., Erban, L., Kooi, H., & Stouthamer, E. (2017). Impacts of 25 years of groundwater extraction on subsidence in the Mekong delta, Vietnam. Environmental Research Letters, 12. https://doi.org/10.1088/1748-9326/aa7146*

*Minderhoud, P. S. J., Middelkoop, H., Erkens, G., & Stouthamer, E. (2020). Groundwater extraction may drown mega-delta : projections of extraction-induced subsidence and elevation of the Mekong delta for the 21st century. Environmental Research Letters. https://doi.org/https://doi.org/10.1088/2515-7620/ab5e21*

*Zoccarato, C., Minderhoud, P. S. J., & Teatini, P. (2018). The role of sedimentation and natural compaction in a prograding delta: insights from the mega Mekong delta, Vietnam. Scientific Reports, 8(1), 1–12. https://doi.org/10.1038/s41598-018-29734-7*

13) Table 4: The caption above S6 should be "Scenarios based on multiple drivers" instead of "individual drivers".

We thank the reviewer for their feedback. We have made the suggested revision

**Reviewer CC1 (Remarks to the Author):**

The paper presents some of the first insights into compound flooding in the Mekong Delta. By providing a range of scenarios, including variations in discharge, storm surge, lowering of the riverbed and integration of (extraction-induced) land subsidence and sea-level rise, the authors aim to attribute modelled inundation to the drivers investigated. They conclude that relative sea-level rise (i.e. land subsidence and absolute sea-level rise) constitutes the most to increase inundation in the delta.

We think this paper addresses a so far huge research gap and can make a significant contribution to the advancement of exposure and risk assessment in the Mekong Delta and the investigation of compound flooding and their local specific characteristics. However, we found some critical aspects that have not been touched by the authors but which are necessary to be at least discussed if they cannot be considered in the processing. We briefly address the major points by listing them as follows and also provide section-wise feedback:

We sincerely thank the reviewer for their time and thoughtful, detailed feedback, which has greatly helped us refine the paper.

**Details and references to the datasets used are missing. The datasets and their quality also need to be reflected in the discussion section.**

1. The assessment heavily relies on the use of a local elevation model but the DEM details are not provided – only the source – how is the DEM acquired, is it a DSM or a DTM? What is vertical accuracy? What are the artifacts (e.g. blocks) in the elevation model that are very visible in the results? What is the influence of these uncertainties/error on the final outcomes of this study? How does the DEM relate to other DEMs/ground truth data on elevation in the delta?

We appreciate the reviewer's thoughtful feedback and recognize the ongoing discussion surrounding the accuracy of DEMs utilized in the VMD. This issue has been highlighted by Minderhoud et al. (2019), who identified substantial inaccuracies in floodplain topography data derived from the Shuttle Radar Topography Mission (SRTM) and MERIT DEM within this region.

In this study, we utilized a Digital Elevation Model (DEM) for the Ca Mau Peninsula acquired from the Ministry of Natural Resources and Environment (MONRE) in 2008, specifically for creating inundation maps in VMD (Tran et al., 2016). This DEM employs the WGS84 coordinate system and the Hon Dau mean sea level vertical datum, with a high spatial resolution of 5 m × 5 m. The DEM was developed using survey points and topographical maps that include elevation points and contour lines derived from geodetic surveys and photogrammetric data (MONRE: Tran et al., 2016). Terrain data were aggregated and selected from topographic maps with the largest scale and best quality to ensure precision (Tran et al., 2016). The DEM represents a Digital Terrain Model (DTM), focusing exclusively on the bare ground surface without any objects such as buildings or vegetation.

Furthermore, the methodology used to develop our selected DEM aligns closely with the approach utilized to create the TOPO DEM described by Minderhoud et al. (2019). This TOPO DEM is acknowledged for its accuracy and reliability in capturing the floodplain topography of the VMD. However, the TOPO DEM, with a coarser grid cell size of 500 × 500 m, is more suitable for broad-scale assessments, such as evaluating the entire VMD. In contrast, our DEM's finer resolution is better suited for localized studies, such as the detailed inundation modelling of the Ca Mau Peninsula. This higher resolution enhances the precision of our analysis, focusing on the region's unique topographical and hydrodynamic characteristics.

We have added information to the line 160, now read:

*A Digital Elevation Model (DEM) for the Ca Mau peninsula, has been acquired from the Ministry of Natural Resources and Environment (MONRE) in 2008 for the purpose of creating inundation maps (Tran et al., 2016) (Appendix A, Fig. A1). This DEM utilizes the WGS84 coordinate system and is based on the Hon Dau means sea level vertical datum, featuring a resolution of 5 m x 5 m. The DEM is using survey points and the topographical maps with the elevation points and contour lines, which were obtained from geodetic survey and photogrammetric data. Terrain data are aggregated and selected from topographic maps of the largest scale and best quality (Tran et al., 2016). The DEM is considered as the best elevation data currently available for the VMD, and it was used for the inundation in VMD under projection of future sea level rise in Vietnam (Tran et al., 2016) as well as other study (Vu et al., 2021; Dang et al., 2023).*

We have, nonetheless, included additional information on this topic in the discussion section.

*Although the accuracy of topographic DEMs in the VMD remains a topic of ongoing discussion (Minderhoud et al., 2019), this study utilized a Digital Elevation Model (DEM) for the Ca Mau Peninsula developed by the Ministry of Natural Resources and Environment (MONRE) in 2008 (Tran et al., 2016). This DEM, with a resolution of 5 m × 5 m, is based on the WGS84 coordinate system and the Hon Dau mean sea level datum, and was created using high-quality survey data, topographic maps, and photogrammetric information (Tran et al., 2016). While this methodology aligns with the TOPO DEM by Minderhoud et al., (2019), which is known for its accuracy at a coarser resolution (500 m × 500 m), the higher resolution of our DEM is better suited for localized analyses of the Ca Mau Peninsula, providing greater precision in capturing the region's topography and inundation dynamics.*

Minderhoud, P. S. J., Coumou, L., Erkens, G., Middelkoop, H., & Stouthamer, E. (2019). Mekong delta much lower than previously assumed in sea-level rise impact assessments. Nature Communications, 1–13. https://doi.org/10.1038/s41467-019-11602-1

Tran, T., Nguyen, V., Huynh, T., Mai, V., Nguyen, X., & HP, D. (2016). Kịch bản biến đổi khí hậu và nước biển dâng cho Việt Nam (Climate change and sea level rise scenarios for Vietnam). Nhà xuất bản Tài Nguyên Môi Trường và Bản Đồ Việt Nam, Bộ Tài Nguyên và Môi Trường, Hà Nội, p 188 [Publishing House of Natural Resources, Enviroment and Cartography, Ministry of Natural Resources and Environment, Ha Noi, p 188]. http://www.imh.ac.vn/files/doc/KichbanBDKH/KBBDKH_2016.pdf

Vu, H. T. D., Trinh, V. C., Tran, D. D., Oberle, P., Hinz, S., & Nestmann, F. (2021). Evaluating the impacts of rice-based protection dykes on floodwater dynamics in the vietnamese mekong delta using geographical impact factor (Gif). Water (Switzerland), 13(9). https://doi.org/10.3390/w13091144

Dang, A. T. N., Reid, M., & Kumar, L. (2023). Coastal Melaleuca wetlands under future climate and sea-level rise scenarios in the Mekong Delta, Vietnam: vulnerability and conservation. Regional Environmental Change, 23(1).

*https://doi.org/10.1007/s10113-022-02009-8*

2. The DEM used was based on data from 2008 – how was this combined with the sea level and the land subsidence data? These should also be computed starting in 2008 – as both processes effectively change the relative elevation of the delta plain.

We appreciate the reviewer's insightful comment regarding the use of DEM data from 2008 and its integration with sea level rise (SLR) and land subsidence data. As discussed in our response to Point 1, high-resolution and up-to-date DEMs for the Mekong Delta are limited. Despite this, the DEM utilized in this study, acquired from the Ministry of Natural Resources and Environment (MONRE) in 2008, is considered one of the most accurate and high-resolution elevation datasets available for the region. Its resolution of 5 m × 5 m and its integration with topographical survey data make it a reliable baseline for inundation modelling (MONRE: Tran et al., 2016).

To address the temporal discrepancy between the DEM (2008) and the 2008 DEM and the 2018-based sea level rise and land subsidence scenario data, we evaluated the changes in these factors from 2008 to 2018 and found them to be relatively small. Specifically, according to Tran et al., 2016 the projected sea level rise in Vietnam's Mekong coastal zone by 2030 is approximately 13 cm compared to the baseline period of 1986–2005. This equates to less than 5 cm of sea level rise between 2008 and 2018, which is negligible for the purposes of this study.

Regarding land subsidence, according to Minderhoud et al., (2017), cumulative land subsidence caused by groundwater extraction from 1991 to 2016 reached a maximum of approximately 30 cm over 25 years in specific areas of the study region (Figure 6). Consequently, it is estimated that between 2008 and 2018, a maximum subsidence of around 15 cm occurred in some areas. Although these changes could introduce slight inaccuracies in the DEM's topography, the variations in relative elevation from 2008 to 2018 remain negligible within the scope of this study. Specifically, (1) sea level rise scenarios are modelled to range between 0.5 m and 1 m, and (2) land subsidence scenarios are represented by values of 0.38 m (σ = 0.21 m) and 1.12 m (σ = 0.59 m), reflecting regional variations. Consequently, the DEM from 2008 remains a suitable baseline for our modelling efforts. By integrating sea level rise and land subsidence projections into our future scenarios, we ensure that the cumulative effects of these processes are accounted for without compromising the validity of the results. This approach provides a reliable framework for assessing inundation risks in the Mekong Delta under various future scenarios.

We have updated our manuscript to acknowledge the methodological limitations (in discussion section), including:

*Additionally, the use of a DEM from 2008 introduces a temporal discrepancy when combined with sea level rise (SLR) and land subsidence scenarios derived from 2018 data. Between 2008 and 2018, SLR in the coastal region was relatively small, with an estimated increase of less than 5 cm according to MONRE projections (Tran et al., 2016), which had minimal impact on the analysis. However, land subsidence during this period was more pronounced, with a maximum of 15 cm in certain areas (Minderhoud et al., 2017). Although these changes could introduce slight inaccuracies in the DEM's*

topography, the variations in relative elevation from 2008 to 2018 remain negligible within the scope of this study. Specifically, sea level rise scenarios range from 0.5 m to 1 m, and land subsidence scenarios range from 0.38 m (σ = 0.21 m) to 1.12 m (σ = 0.59 m), reflecting regional variability. Thus, the relative elevation changes from 2008 to 2018 are considered minimal within the scope of our analysis.

* Tran, T., Nguyen, V., Huynh, T., Mai, V., Nguyen, X., & HP, D. (2016). Kịch bản biến đổi khí hậu và nước biển dâng cho Việt Nam (Climate change and sea level rise scenarios for Vietnam). Nhà xuất bản Tài Nguyên Môi Trường và Bản Đồ Việt Nam, Bộ Tài Nguyên và Môi Trường, Hà Nội, p 188 [Publishing House of Natural Resources, Enviroment and Cartography, Ministry of Natural Resources and Environment, Ha Noi, p 188]. http://www.imh.ac.vn/files/doc/KichbanBDKH/KBBDKH_2016.pdf

Minderhoud, P. S. J., Erkens, G., Pham, V. H., Bui, V. T., Erban, L., Kooi, H., & Stouthamer, E. (2017). Impacts of 25 years of groundwater extraction on subsidence in the Mekong delta, Vietnam. Environmental Research Letters, 12. https://doi.org/10.1088/1748-9326/aa7146

3. Reference for the gauge data is missing and should be provided.

We thank the reviewer for their feedback. We have updated the information in line 155, which now reads:

"The dataset comprises a single set of long-term daily water flow data at Katie, obtained from the Mekong River Commission (MRC: https://portal.mrcmekong.org/monitoring/river-monitoring-telemetry). Other data, including water discharge and water levels in the mainstream of the VMD, are recorded hourly, with daily maximum and minimum water levels collected from the Ca Mau gauge within the study area. These measurements are obtained from gauges operated by the Southern Regional Hydro-Meteorological Center (SRHMC) with water levels referenced to the Hon Dau Mean Sea Level (MSL) datum and provided by SIWRR (http://www.siwrr.org.vn/)".

4. LULC details / references are not provided and should be briefly summarised or at least provided by adding a reference.

We appreciate the reviewer's feedback and have updated the caption of Figure 2, line 246, as follows:

Figure 2. (a) Location of the study area and land use data for the year 2022 (The People's Committee of Ca Mau Province, 2023); (b) the 1D model used in this study.

5. Details on roughness are missing. Was it a single value? Were the initial roughness value(s) determined based on the land-use land cover map? Please provide some more information.

We appreciate the reviewer's comment. As referenced in point 2, RC1, we have added the following information to line 182:

The initial roughness values were based on Manh et al., (2014), who categorized the model domain into distinct zones, assigning specific Manning's n values to each zone to represent their unique hydraulic characteristics. These values were systematically refined to achieve the best agreement between the model's predicted water levels and discharges and the observed data. Specifically, the calibration process utilized monitoring stations located along the main Mekong channel in Vietnam (Fig. 2b, Table 1). Model outputs were compared with observed water levels and discharges, and any

*discrepancies prompted adjustments to the zone-specific roughness values. The model was iteratively rerun until optimal results were achieved. The final calibrated Manning's n values are detailed in Table 3.*

**Table 3.** *The Manning roughness coefficient (n) is categorized based on different zones using in the modelling. The Manning roughness in the channel changing for different cross-sections within the same zone*

| Zone | Description | Manning's coefficient (n) |
|------|-------------|---------------------------|
| 1 | Mekong River: Kratie to Phnom Penh | 0.032 |
| 2 | Mekong River: Phnom Penh to Tan Chau | 0.032 to 0.027 |
| 3 | Mekong River: Tan Chau to My Thuan | 0.027 to 0.025 |
| 4 | Mekong River: My Thuan to River months | 0.025 to 0.016 |
| 5 | Bassac River: Phnom Penh to Chau Doc | 0.032 |
| 6 | Bassac River: Chau Doc to Can Tho | 0.032 to 0.025 |
| 7 | Bassac River: Can Tho to River months | 0.025 to 0.017 |
| 8 | Side channels | 0.033 |
| 9 | Floodplain | 0.033 |

6. The paper uses a quantification of 'Delta subsidence' while based from the description this is based on a groundwater model however the paper doesn't reference the original model source (i.e. Minderhoud et al., 2017), which is required as this holds the assumption and also shortcomings of the modelling approach. It is important to note that the modelling results only provide land subsidence following groundwater extraction and not includes other components of land subsidence – this should be made explicit in the result presentation (e.g. change delta subsidence to extraction-induced subsidence) and discussion on the other components not included in the assessment. For this also other literature specifically on land subsidence in Ca Mau on the other components and local measurements can be used: e.g. Zoccarato et al., 2018 (10.1038/s41598-018-29734-7), Karlsrud et al., 2020 (10.5194/piahs-382-111-2020), de Wit et al., 2021 (10.3390/rs13020189). Also Nguyen et al., 2023 (10.1016/j.ecss.2023.108259) provides a valuable comparison with water level measurements to discussed in this study.

We appreciate the reviewer for their insightful comment. In response, we have made adjustments to our land subsidence scenarios. The revised version of line 271 now reads:

***Delta Subsidence (S4_a, S4_b):*** *These scenarios evaluate the changes in inundation associated with land subsidence in the future. In this study, we only assess the impact of groundwater extraction and do not include other components of land subsidence and does not include other contributing factors such as natural subsidence, tectonic movements, or other human activities (Minderhoud et al., 2017; Zoccarato et al., 2018); Karlsrud et al., 2020). This focus is due to the fact that land subsidence in the Ca Mau Peninsula is primarily driven by groundwater extraction (Minderhoud et al., 2017; Karlsrud et al., 2020).*

The spatial map depicting future land subsidence within the VMD is derived from Minderhoud et al., (2020). This dataset includes land subsidence scenarios labeled B1.5 for the year 2050 ($B1.5_{2050}$) and B2 for the year 2100 ($B2_{2100}$). The B1.5 scenario simulates land subsidence under a moderate increase in groundwater extraction, serving as a pathway between the B1 scenario (moderate increase) and the B2 scenario (extreme increase). Both scenarios follow a non-mitigation pathway, where groundwater extraction continues to increase without any reduction measures, leading to greater land subsidence over time Minderhoud et al., (2020). In this study, we use the land subsidence scenarios $B1.5_{2050}$ and $B2_{2100}$, which are referred to as the S4_a and S4_b scenarios, respectively. The S4_a scenario represents an average land subsidence of 0.38 m (σ = 0.21 m), while the S4_b scenario reflects 1.12 m (σ = 0.59 m) of subsidence in the Ca Mau Peninsula.

Additionally, we propose including this limitation in the discussion section.

Regarding land subsidence scenarios, this study focuses solely on the impact of groundwater extraction, excluding other factors contributing to land subsidence, such as natural subsidence, tectonic movements, or other human activities (Minderhoud et al., 2017; Zoccarato et al., 2018); Karlsrud et al., 2020). This emphasis is based on the fact that groundwater extraction is the primary driver of land subsidence in the Ca Mau Peninsula (Minderhoud et al., 2017; Karlsrud et al., 2020).

However, it is important to acknowledge that reduced sediment input from upstream could exacerbate natural subsidence processes, particularly through natural compaction, as discussed by Zoccarato et al., (2018). Therefore, while this study does not account for these additional subsidence factors, the broader context of sediment deficits should be considered as a contributing factor to the overall land subsidence in the region.

*Minderhoud, P. S. J., Erkens, G., Pham, V. H., Bui, V. T., Erban, L., Kooi, H., & Stouthamer, E. (2017). Impacts of 25 years of groundwater extraction on subsidence in the Mekong delta, Vietnam. Environmental Research Letters, 12. https://doi.org/10.1088/1748-9326/aa7146

Minderhoud, P. S. J., Middelkoop, H., Erkens, G., & Stouthamer, E. (2020). Groundwater extraction may drown mega-delta : projections of extraction-induced subsidence and elevation of the Mekong delta for the 21st century. Environmental Research Letters. https://doi.org/https://doi.org/10.1088/2515-7620/ab5e21

Zoccarato, C., Minderhoud, P. S. J., & Teatini, P. (2018). The role of sedimentation and natural compaction in a prograding delta: insights from the mega Mekong delta, Vietnam. Scientific Reports, 8(1), 1–12. https://doi.org/10.1038/s41598-018-29734-7

Karlsrud, K., Tunbridge, L., Khanh, N. Q., & Dinh, N. Q. (2020). Preliminary results of land subsidence monitoring in the Ca Mau Province. Proceedings of the International Association of Hydrological Sciences, 382, 111–115. https://doi.org/10.5194/piahs-382-111-2020

7. Coastal erosion and shoreline change are included in discussion while happening in the coastal zone – especially with inundations projected up to 2 meter. Please include this in the discussion (e.g. Anthony et al., 2015)

We thank the reviewer for their valuable and thoughtful comment. We have incorporated this suggestion into the discussion, and line 447 now reads as follows:

Although land subsidence and SLR contribute to elevation loss and increased inundation, they may also create conditions favourable for sediment deposition in flooded areas. This sediment trapping

mechanism has the potential to partially offset elevation loss over time, thereby influencing future inundation depths. However, the effectiveness of this natural compensatory process is significantly constrained by the sharp reduction in sediment flow from upstream rivers, largely due to damming (Kondolf et al., 2014). These dams have resulted in a projected 57% decrease in suspended sediment flux, from 99 Mt yr$^{-1}$ (1980-2009) to 43 Mt yr$^{-1}$ by 2020-2029 (Bussi et al. 2021). Sediment transport is crucial for maintaining coastal stability, replenishing eroded areas, and preserving elevation. The decline in sediment supply, driven by upstream dams and other human activities, has worsened the region's vulnerability, accelerating coastal erosion and shoreline retreat (Anthony et al., 2015; Tu et al., 2019).

*Anthony, E. J., Brunier, G., Besset, M., Goichot, M., Dussouillez, P., & Nguyen, V. L. (2015). Linking rapid erosion of the Mekong River delta to human activities. Scientific Reports, 5. https://doi.org/10.1038/srep14745

Bussi, G., Darby, S. E., Whitehead, P. G., Jin, L., Dadson, S. J., Voepel, H. E., Vasilopoulos, G., Hackney, C. R., Hutton, C., Berchoux, T., Parsons, D. R., & Nicholas, A. (2021). Impact of dams and climate change on suspended sediment flux to the Mekong delta. Science of the Total Environment, 755. https://doi.org/10.1016/j.scitotenv.2020.142468

Kondolf, G. M., Rubin, Z. K., & Minear, J. T. (2014). Dams on the Mekong: Cumulative sediment starvation. Water Resources Research, 50, 5158–5169. https://doi.org/10.1002/2013WR014651

Tu, L. X., Thanh, V. Q., Reyns, J., Van, S. P., Anh, D. T., Dang, T. D., & Roelvink, D. (2019). Sediment transport and morphodynamical modeling on the estuaries and coastal zone of the Vietnamese Mekong Delta. Continental Shelf Research, 64–76. https://doi.org/10.1016/j.csr.2019.07.015

8. In addition to elevation loss following land subsidence and SLR – the accretion of sediment should be discussed – as this may cause a positive elevation change. Especially with high flood/inundation with water - sedimentation dynamics will change and therefore elevation – which in turn changes the inundation depth. The lead author has experience in flood plain sedimentation and adding these dynamics in the discussion would strengthen the changes and future dynamics the paper aims to evaluate.

We appreciate the reviewer for their insightful and constructive comment. Please refer to the response in point 7 above.

9. The discussion section is very short and in its current version is missing several aspects that require critical reflection. We suggest to insert a subsection where the quality of underlying datasets as well as the limitations are discussed. And include a second subsection to discuss implications of your modelling results for the delta's future and its development/management. Only a very limited number of mitigation and adaptation measures is discussed (i.e. tidal barriers) and these are hardly suitable to tackle the causes and drivers of compound flooding as well as sustaining mangrove ecosystems. Thus can be improved by restricting the discussion.

We appreciate the reviewer's insightful and constructive comment. We have incorporated this suggestion into the discussion section and expanded the limitations within this section. It now reads:

*4 Discussion*

*The aforementioned findings underscore the potential anthropogenic impact on the evolution of the inundation regime across the Ca Mau peninsula. These results suggest that the influences of riverbed lowering in the primary Mekong channel, and the occurrence of storm surges along the mainstream*

Mekong River are unlikely to significantly affect the inundation regime in this region, while high-water discharge did cause the inundation in this study area to rise only by approximately 0.1-0.3 m. The minimal impacts can be attributed to the prominence of high-water discharge, riverbed lowering, and storm surges primarily occurring in the upstream region and in the main Mekong and Bassac channels. Since the study area is situated farther from these channels and closer to the sea on both sides, these impacts tend to diminish. However, the results emphasizes that land subsidence and tidal influence combined with SLR will be the primary factors driving the escalation of the inundation regime in the Ca Mau peninsula in the future, both in extent and magnitude. In the projected future by the mid-century, under the S6 scenarios, a combination of high-water discharge, storm surge, land subsidence, and sea level rise are projected to inundate 96% of the Ca Mau Peninsula, with an average inundation depth of 0.91 m (σ = 0.44 m). Looking towards the end of the century under the S7 scenarios, the compounded effects of high-water discharge, storm surge, land subsidence, and a sea level rise are expected to inundate 99% of the Ca Mau Peninsula, with an average inundation depth of 2.20 m (σ = 0.68 m). However, it should be noted that these projections are based on the SSP5-8.5 scenarios, which represent the very worst case with low confidence. The ramifications of this increasing trend in inundation are now deliberated in relation to the region's future sustainability and its capacity to adapt to present and future transformations.

The increasing inundation resulting from land subsidence and SLR poses a significant threat of saltwater intrusion into inland regions, which will have severe consequences for agriculture, aquaculture, and infrastructure in the Ca Mau Peninsula. This area is not only economically important but also home to one of the most diverse natural ecosystems in the Mekong Delta, with over 79,100 hectares of mangrove forests that play a vital role in maintaining the region's ecological balance. These mangrove forests are essential for protecting coastal zones, regulating freshwater flow, and supporting local biodiversity. However, recent years have seen significant anthropogenic pressure on this ecosystem, particularly from the development of transportation infrastructure, irrigation systems, and water control structures. These developments have led to substantial degradation of the mangrove forests (Son et al., 2015; Hauser et al., 2020; Thuy et al., 2023), further compounded by the potential rise in inundation levels driven by land subsidence and SLR. This combination of factors threatens to accelerate the loss of these critical ecosystems, undermining their ability to provide natural coastal defence, support biodiversity, and mitigate the effects of climate change (Barbier et al., 2008; Temmerman et al., 2013; Jones et al. 2020; Sunkur et al., 2023).

Although land subsidence and SLR contribute to elevation loss and increased inundation, they may also create conditions favourable for sediment deposition in flooded areas. This sediment trapping mechanism has the potential to partially offset elevation loss over time, thereby influencing future inundation depths. However, the effectiveness of this natural compensatory process is significantly constrained by the sharp reduction in sediment flow from upstream rivers, largely due to damming (Kondolf et al., 2014). These dams have resulted in a projected 57% decrease in suspended sediment flux, from 99 Mt yr$^{-1}$ (1980-2009) to 43 Mt yr$^{-1}$ by 2020-2029 (Bussi et al., 2021). Sediment transport is crucial for maintaining coastal stability, replenishing eroded areas, and preserving elevation. The

*decline in sediment supply, driven by upstream dams and other human activities, has worsened the region's vulnerability, accelerating coastal erosion and shoreline retreat (Anthony et al., 2015; Tu et al., 2019).*

*In addition, the study also reveals concerns regarding the region's infrastructure, particularly road construction. According to the national North-South Expressway's design specifications for the period 2021-2025 (No: 912/QĐ-BGTVT), the road crest is set 2.2 m above mean sea level in the Ca Mau area (Fig. 2). However, land subsidence projections of approximately 0.77 m by mid-century and 2.34 m by the end of the century could lower the road crest to 1.43 m and -0.14 m, respectively. These reductions may not ensure adequate protection, especially considering predicted maximum water levels of approximately 1.59 m by 2050 and 2.20 m by 2100. This assessment is based on Vietnam's road construction standards (TCVN 4054: 2005). Furthermore, the development of infrastructure contributes additional pressure to the deltaic land surface, exacerbating land subsidence in these areas.*

*To mitigate the impacts of flooding caused by SLR and land subsidence in the Ca Mau Peninsula, it is crucial to implement a combination of sustainable strategies that address both the natural and anthropogenic drivers of vulnerability. A primary focus must be on reducing groundwater extraction, the leading cause of land subsidence in the region. This is critical not only for mitigating land subsidence but also for enhancing the area's resilience to the escalating impacts of climate change (Minderhoud et al., 2020). Additionally, increasing sediment flow from upstream rivers is essential for both replenishing eroded coastal areas and counteracting the effects of land subsidence. By improving sediment management in river systems, such as restoring natural sediment transport pathways and removing barriers like dams, the flow of sediment can be enhanced. Moreover, implementing water-saving technologies, such as efficient irrigation systems and the upgrading of water channels from the Bassac River, can significantly reduce the demand for groundwater in agriculture and domestic use. Moreover, developing alternative water sources, such as, rainwater harvesting, desalination, and surface water reservoirs, which can reduce dependency on groundwater, ensuring reliable water supplies during periods of drought or high demand.*

*In parallel, restoring and protecting the natural balance between saltwater and freshwater systems is essential. The creation or rehabilitation of tidal marshes and mangrove forests along the coastline can serve as natural barriers, filtering salinity, protecting against erosion and storm surges, and providing critical habitat for biodiversity while sustaining local livelihoods (Barbier et al., 2008;Temmerman et al., 2013; Jones et al., 2020; Sunkur et al., 2023; Dominicis et al., 2023). In addition, the integration of engineered infrastructure, including tidal barriers, levees, and dikes, can help prevent saltwater intrusion and manage flood risks, particularly in areas most vulnerable to the effects of SLR.*

*Community engagement is equally important in ensuring the long-term success of these strategies. Raising awareness about the negative impacts of excessive groundwater extraction and the benefits of sustainable water use can foster local participation in conservation efforts. Educational campaigns and training programs can also empower local stakeholders to adopt water-saving practices, contributing to the sustainable management of this vital resource. Finally, investing in robust*

monitoring systems is essential for tracking groundwater levels, land subsidence rates, and water usage patterns. Accurate, real-time data can inform effective decision-making, enabling authorities to prioritize interventions and manage resources efficiently. By integrating these natural, infrastructural, and community-based strategies, the Ca Mau Peninsula can build resilience against flooding, safeguard its ecosystems, and ensure the sustainability of its agricultural and aquacultural industries, even in the face of rising seas and land subsidence.

In this study, flood risk in the Ca Mau Peninsula was assessed using 1D modelling, taking into account both individual and combined factors. Although the results provide useful insights, it is essential to recognize the inherent uncertainties associated with the model scenarios and methodologies used.

One limitation arises from the use of a one-dimensional model, which calculates water levels in the floodplain through spatial interpolation from the water levels in surrounding channels. While this approach may be less accurate compared to a 2D model, it still provides a reasonable estimate when combined with 2D modelling. Specifically, the 1D model utilized in this study incorporates extensive data on secondary channels within the Vietnam Mekong Delta (VMD), particularly within the Ca Mau Peninsula. The secondary channel data resolution ranges from 1 km to 3 km per channel in the study area, ensuring the model accurately represents the region's complex hydrodynamics. Moreover, the model includes approximately 4,235 river branches and 564 floodplain compartments across the VMD, as detailed by Dung et al., (2011). This configuration allows the model to effectively simulate dynamic water exchanges between the mainstream river and the floodplain system, including water level patterns within the floodplain, thus improving the representation of local hydrological conditions despite the limitations of the 1D approach. In addition, the model does not account for long-term sediment transport, which could influence changes in the floodplain area, with sedimentation potentially counteracting land subsidence (Hung et al., 2014; Zoccarato et al., 2018). While this limitation is acknowledged, the study partially addresses it by incorporating predicted bathymetric changes in the main Mekong River up to 2038, considering factors such as sediment deficits and sand mining activities (Vasilopoulos et al., 2021). These adjustments are intended to capture certain aspects of sediment dynamics and their potential impact on the delta.

Although the accuracy of topographic DEMs in the VMD remains a topic of ongoing discussion (Minderhoud et al., 2019), this study utilized a Digital Elevation Model (DEM) for the Ca Mau Peninsula developed by the Ministry of Natural Resources and Environment (MONRE) in 2008 (Tran et al., 2016). This DEM, with a resolution of 5 m × 5 m, is based on the WGS84 coordinate system and the Hon Dau mean sea level datum, and was created using high-quality survey data, topographic maps, and photogrammetric information (Tran et al., 2016). While this methodology aligns with the TOPO DEM by Minderhoud et al., (2019), which is known for its accuracy at a coarser resolution (500 m × 500 m), the higher resolution of our DEM is better suited for localized analyses of the Ca Mau Peninsula, providing greater precision in capturing the region's topography and inundation dynamics.

The use of a 5x5 m resolution DEM in this study has provided a detailed representation of the topographical variations within the study area. However, this high level of detail also introduces certain

artefacts that become apparent in Figure 6 and 7. These artefacts stem from the granularity of the DEM dataset, which captures subtle variations in elevation between adjacent cells. Such elevation differences are particularly pronounced in the mapping of inundation levels, where thresholds such as <0.1 m (unflooded; white colour, Fig. 6) and 0.1–0.4 m (flooded; green colour, Fig. 6) highlight the varying extents of inundated areas. While this granularity allows for a more precise delineation of inundation zones, it also introduces abrupt transitions in the inundation extent that may not accurately reflect real-world conditions. These artefacts are especially noticeable in flat or low-lying areas, where small discrepancies in elevation can disproportionately affect the modelled inundation extent. Despite these limitations, the 5x5 m DEM provides valuable insights into the spatial distribution of inundation areas at a resolution suitable for regional-scale assessments. The artefacts observed in this study highlight the importance of balancing DEM resolution with the study's scale and objectives to ensure accurate and reliable outcomes.

The assumption that areas with a depth of less than 0.1 m are classified as unflooded is typically used to identify negligible inundation. However, in urban areas, even shallow water depths can become hazardous when associated with high flow velocities. Such conditions can lead to risks, such as infrastructure damage, heightened erosion, and potential threats to public safety.

Additionally, the use of a DEM from 2008 introduces a temporal discrepancy when combined with sea level rise (SLR) and land subsidence scenarios derived from 2018 data. Between 2008 and 2018, SLR in the coastal region was relatively small, with an estimated increase of less than 5 cm according to MONRE projections (Tran et al., 2016), which had minimal impact on the analysis. However, land subsidence during this period was more pronounced, with a maximum of 15 cm in certain areas (Minderhoud et al., 2017). Although these changes could introduce slight inaccuracies in the DEM's topography, the variations in relative elevation from 2008 to 2018 remain negligible within the scope of this study. Specifically, sea level rise scenarios range from 0.5 m to 1 m, and land subsidence scenarios range from 0.38 m ($\sigma$ = 0.21 m) to 1.12 m ($\sigma$ = 0.59 m), reflecting regional variability. Thus, the relative elevation changes from 2008 to 2018 are considered minimal within the scope of our analysis.

Regarding land subsidence scenarios, this study focuses solely on the impact of groundwater extraction, excluding other factors contributing to land subsidence, such as natural subsidence, tectonic movements, or other human activities (Minderhoud et al., 2017; Zoccarato et al., 2018; Karlsrud et al., 2020). This emphasis is based on the fact that groundwater extraction is the primary driver of land subsidence in the Ca Mau Peninsula (Minderhoud et al., 2017;  Karlsrud et al., 2020).

However, it is important to acknowledge that reduced sediment input from upstream could exacerbate natural subsidence processes, particularly through natural compaction, as discussed by Zoccarato et al., (2018). Therefore, while this study does not account for these additional subsidence factors, the broader context of sediment deficits should be considered as a contributing factor to the overall land subsidence in the region.

Finally, the inundation scenarios in this study are based on long-term projections that consider the cumulative impacts of land subsidence and sea level rise up to 2100. While these projections are essential for evaluating future risks and guiding long-term planning, they may not fully account for short-term fluctuations or temporary environmental changes, potentially leading to an overestimation of short-term impacts. Although the focus on long-term trends offers valuable insights into future scenarios, further research that includes short-term variability is needed to provide a more comprehensive understanding of the region's vulnerability to flooding and land subsidence in the near term.

*Anthony, E. J., Brunier, G., Besset, M., Goichot, M., Dussouillez, P., & Nguyen, V. L. (2015). Linking rapid erosion of the Mekong River delta to human activities. Scientific Reports, 5. https://doi.org/10.1038/srep14745

Barbier, E. B., Koch, E. W., Silliman, B. R., Hacker, S. D., Wolanski, E., Primavera, J., Granek, E. F., Polasky, S., Aswani, S., Cramer, L. A., Stoms, D. M., Kennedy, C. J., Bael, D., Kappel, C. V., Perillo, G. M. E., & Reed, D. J. (2008). Coastal ecosystem-based management with nonlinear ecological functions and values. Science, 319(5861), 321–323. https://doi.org/10.1126/science.1150349

Bussi, G., Darby, S. E., Whitehead, P. G., Jin, L., Dadson, S. J., Voepel, H. E., Vasilopoulos, G., Hackney, C. R., Hutton, C., Berchoux, T., Parsons, D. R., & Nicholas, A. (2021). Impact of dams and climate change on suspended sediment flux to the Mekong delta. Science of the Total Environment, 755. https://doi.org/10.1016/j.scitotenv.2020.142468

De Dominicis, M., Wolf, J., van Hespen, R., Zheng, P., & Hu, Z. (2023). Mangrove forests can be an effective coastal defence in the Pearl River Delta, China. Communications Earth and Environment, 4(1). https://doi.org/10.1038/s43247-022-00672-7

Dung, N. V., Merz, B., Bárdossy, A., Thang, T. D., & Apel, H. (2011). Multi-objective automatic calibration of hydrodynamic models utilizing inundation maps and gauge data. Hydrology and Earth System Sciences, 15, 1339–1354. https://doi.org/10.5194/hess-15-1339-2011

Hauser, L. T., Hauser, L. T., Binh, N. A., Hoa, P. V., Quan, N. H., & Timmermans, J. (2020). Gap-Free Monitoring of Annual Mangrove Forest Dynamics in Ca Mau Province, Vietnamese Mekong Delta, Using the Landsat-7-8 Archives and Post-Classification Temporal Optimization. MDPI. https://doi.org/doi:10.3390/rs12223729

Hung, N. N., Delgado, J. M., Güntner, A., Merz, B., Bárdossy, A., & Apel, H. (2014). Sedimentation in the fl oodplains of the Mekong Delta, Vietnam. Part I : suspended sediment dynamics. 3144, 3132–3144. https://doi.org/10.1002/hyp.9856

Jones, H. P., Nickel, B., Srebotnjak, T., Turner, W., Gonzalez-Roglich, M., Zavaleta, E., & Hole, D. G. (2020). Global hotspots for coastal ecosystem-based adaptation. PLoS ONE, 15(5), 1–17. https://doi.org/10.1371/journal.pone.0233005

Karlsrud, K., Tunbridge, L., Khanh, N. Q., & Dinh, N. Q. (2020). Preliminary results of land subsidence monitoring in the Ca Mau Province. Proceedings of the International Association of Hydrological Sciences, 382, 111–115. https://doi.org/10.5194/piahs-382-111-2020

Kondolf, G. M., Rubin, Z. K., & Minear, J. T. (2014). Dams on the Mekong: Cumulative sediment starvation. Water Resources Research, 50, 5158–5169. https://doi.org/10.1002/2013WR014651

Minderhoud, P. S. J., Coumou, L., Erkens, G., Middelkoop, H., & Stouthamer, E. (2019). Mekong delta much lower than previously assumed in sea-level rise impact assessments. Nature Communications, 1–13. https://doi.org/10.1038/s41467-019-11602-1

Minderhoud, P. S. J., Erkens, G., Pham, V. H., Bui, V. T., Erban, L., Kooi, H., & Stouthamer, E. (2017). Impacts of 25 years of groundwater extraction on subsidence in the Mekong delta, Vietnam. Environmental Research Letters, 12. https://doi.org/10.1088/1748-9326/aa7146

Minderhoud, P. S. J., Middelkoop, H., Erkens, G., & Stouthamer, E. (2020). Groundwater extraction may drown mega-delta : projections of extraction-induced subsidence and elevation of the Mekong delta for the 21st century. Environmental Research Letters. https://doi.org/https://doi.org/10.1088/2515-7620/ab5e21

Son, N., Chen, C., Chang, N., Chen, C., Chang, L., & Thanh, B. (2015). Mangrove Mapping and Change Detection in Ca Mau Peninsula, Vietnam, Using Landsat Data and Object-Based Image Analysis. IEEE Journal of Selected Topics in Applied Earth Observations and Remote Sensing, 8(2), 503–510. https://doi.org/10.1109/JSTARS.2014.2360691

Sunkur, R., Kantamaneni, K., Bokhoree, C., & Ravan, S. (2023). Mangroves' role in supporting ecosystem-based techniques to reduce disaster risk and adapt to climate change: A review. Journal of Sea Research, 196(October), 102449. https://doi.org/10.1016/j.seares.2023.102449

Temmerman, S., Meire, P., Bouma, T. J., Herman, P. M. J., Ysebaert, T., & De Vriend, H. J. (2013). Ecosystem-based coastal defence in the face of global change. Nature, 504(7478), 79–83. https://doi.org/10.1038/nature12859

Thuy, L., Nguyen, M., Thi, H., Choi, E., & Sun, P. (2023). Estuarine, Coastal and Shelf Science Distribution of mangroves with different aerial root morphologies at accretion and erosion sites in Ca Mau Province, Vietnam. Estuarine, Coastal and

Shelf Science, 287(March), 108324. https://doi.org/10.1016/j.ecss.2023.108324

Tran, T., Nguyen, V., Huynh, T., Mai, V., Nguyen, X., & HP, D. (2016). Kịch bản biến đổi khí hậu và nước biển dâng cho Việt Nam *(Climate change and sea level rise scenarios for Vietnam). Nhà xuất bản Tài Nguyên Môi Trường và Bản Đồ Việt Nam, Bộ Tài Nguyên và Môi Trường, Hà Nội, p 188 [Publishing House of Natural Resources, Enviroment and Cartography, Ministry of Natural Resources and Environment, Ha Noi, p 188].* http://www.imh.ac.vn/files/doc/KichbanBDKH/KBBDKH_2016.pdf

Tu, L. X., Thanh, V. Q., Reyns, J., Van, S. P., Anh, D. T., Dang, T. D., & Roelvink, D. (2019). Sediment transport and morphodynamical modeling on the estuaries and coastal zone of the Vietnamese Mekong Delta. Continental Shelf Research, 64–76. https://doi.org/10.1016/j.csr.2019.07.015

Vasilopoulos, G., Quan, Q. L., Parsons, D. R., Darby, S. E., Tri, V. P. D., Hung, N. N., Haigh, I. D., Voepel, H. E., Nicholas, A. P., & Aalto, R. (2021). Establishing sustainable sediment budgets is critical for climate-resilient mega-deltas. Environmental Research Letters, 16. https://doi.org/10.1088/1748-9326/ac06fc

Zoccarato, C., Minderhoud, P. S. J., & Teatini, P. (2018). The role of sedimentation and natural compaction in a prograding delta: insights from the mega Mekong delta, Vietnam. Scientific Reports, 8(1), 1–12. https://doi.org/10.1038/s41598-018-29734-7

Minor comments:

**Introduction**

- 18: there is no "natural disaster", please delete "natural".

We have made the edit

- 40-41: "Sand dredging rise" sounds a bit strange. Replace it by "increased sand mining".

We have made the edit

- 70: "Vertical sinking" sounds a bit strange. Remove "vertical".

We have made the edit

- 74: Please cite and acknowledge the required source publications of the NASA SLR projection tool, i.e. Fox-Kemper et al. (2021) and the data works of Garner et al. Please check this throughout the entire manuscript.

We have made the edit

- 86-90: Please provide references.

We have included the references.

- 93: Vulnerability has not been adequately addressed so far. The previous paragraphs were all about future trends and increasing exposure. Please insert a few lines that underpin your statement of increased vulnerability (i.e., studies focussing on societal aspects) or do not call it vulnerability.

We have made the edit, the line 93 now read:

*Recognizing the profound significance of this inquiry, particularly in relation to the heightened flood risks faced by local communities, our overarching aim is to refine decision-making frameworks, enhance flood management strategies, and bolster regional planning initiatives*

- 99-101: Can you provide a reference who will consider it? MONRE, etc.?

We appreciate the reviewer for their valuable feedback, we have made the edit that the line 99-101 now read:

*Therefore, this article represents an important contribution for analysis, forecasting, warning, and proposing various flood control solutions for the Ca Mau region. Based on future predictions, water control systems and future developments will be planned, including expressways, coastal embankments to control tidal surges in the West Sea, East Sea, and saltwater control gates along the Bassac river, which are essential for authorities like MONRE, MOT, MOC, MARD, and MPI in making informed long-term development decisions. The successful implementation of these measures will strengthen the region's flood resilience and support sustainable growth amid climate change. The article aims to achieve the following objectives:*

- 111: There is no reference made to figure 1 in the text. Either integrate it into the text (for example if you use it to validate your model) or remove it.

We have made the edit and integrated it into the text in line 98.

**Methods**

- 129-130: Similar as reference is provided for monsoon precipitation, please also provide a reference when information on elevation above mean sea level is given.

We have made the edit to include the reference (MRC, 2005) for monsoon precipitation and (Karlsrud et al., 2020; Tran et al., 2024) for elevation data relevant to the study region.

*MRC. (2005). Overview of the Hydrology of the Mekong Basin. November.*
*Karlsrud, K., Tunbridge, L., Khanh, N. Q., & Dinh, N. Q. (2020). Preliminary results of land subsidence monitoring in the Ca Mau Province. Proceedings of the International Association of Hydrological Sciences, 382, 111–115. https://doi.org/10.5194/piahs-382-111-2020*
*Tran, A. Van, Brovelli, M. A., Ha, K. T., Khuc, D. T., Tran, D. N., Tran, H. H., & Le, N. T. (2024). Land Subsidence Susceptibility Mapping in Ca Mau Province, Vietnam, Using Boosting Models. ISPRS International Journal of Geo-Information, 13(5), 1–24. https://doi.org/10.3390/ijgi13050161*

- 145: Provide a reference for the land-use data used.

We have made the edit to include the reference (The People's Committee of Ca Mau Province, 2023);

*\*The People's Committee of Ca Mau Province (2023). Comprehensive report on planning for Ca Mau Province during 2021–2030, with a vision to 2050. Ca Mau Province.*

- 158-159: Provide year of datum establishment in brackets or provide reference where this information is entailed.

We have made the edit, the line 154-159 now read *"The dataset comprises a single set of long-term daily water flow data at Katie, obtained from the Mekong River Commission (MRC: https://portal.mrcmekong.org/monitoring/river-monitoring-telemetry). Other data, including water discharge and water levels in the mainstream of the VMD, are recorded hourly, with daily maximum and minimum water levels collected from the Ca Mau gauge within the study area. These measurements are obtained from gauges operated by the Southern Regional Hydro-Meteorological Center (SRHMC) with water levels referenced to the Hon Dau Mean Sea Level (MSL) datum and provided by SIWRR (http://www.siwrr.org.vn/)".*

- 159-162: More details on the DEM are needed. How was it acquired? Airborne LiDAR? Is it a DSM or a DTM? This information is very important as it has crucial impacts on your model results.

We appreciate the reviewer's thoughtful feedback, refer to response to point 1, we have added information to line 160, which now reads:

*A Digital Elevation Model (DEM) for the Ca Mau peninsula, has been acquired from the Ministry of Natural Resources and Environment (MONRE) in 2008 for the purpose of creating inundation maps (Tran et al., 2016) (Appendix A, Fig. A1). This DEM utilizes the WGS84 coordinate system and is based on the Hon Dau means sea level vertical datum, featuring a resolution of 5 m x 5 m. The DEM is using survey points and the topographical maps with the elevation points and contour lines, which were obtained from geodetic survey and photogrammetric data. Terrain data are aggregated and selected from topographic maps of the largest scale and best quality (Tran et al., 2016). The DEM is considered as the best elevation data currently available for the VMD, and it was used for the inundation in VMD under projection of future sea level rise in Vietnam (Tran et al., 2016) as well as other study (Vu et al. 2021; Dang et al., 2023).*

*Tran, T., Nguyen, V., Huynh, T., Mai, V., Nguyen, X., & HP, D. (2016). Kịch bản biến đổi khí hậu và nước biển dâng cho Việt Nam (Climate change and sea level rise scenarios for Vietnam). Nhà xuất bản Tài Nguyên Môi Trường và Bản Đồ Việt Nam, Bộ Tài Nguyên và Môi Trường, Hà Nội, p 188 [Publishing House of Natural Resources, Enviroment and Cartography, Ministry of Natural Resources and Environment, Ha Noi, p 188]. http://www.imh.ac.vn/files/doc/KichbanBDKH/KBBDKH_2016.pdf*

*Vu, H. T. D., Trinh, V. C., Tran, D. D., Oberle, P., Hinz, S., & Nestmann, F. (2021). Evaluating the impacts of rice-based protection dykes on floodwater dynamics in the vietnamese mekong delta using geographical impact factor (Gif). Water (Switzerland), 13(9). https://doi.org/10.3390/w13091144*

*Dang, A. T. N., Reid, M., & Kumar, L. (2023). Coastal Melaleuca wetlands under future climate and sea-level rise scenarios in the Mekong Delta, Vietnam: vulnerability and conservation. Regional Environmental Change, 23(1). https://doi.org/10.1007/s10113-022-02009-8*

- 163-164: Provide a reference for the gauge data used.

We appreciate the reviewer's feedback. The information has been updated in line 155, which now reads:

*"The dataset comprises a single set of long-term daily water flow data at Katie, obtained from the Mekong River Commission (MRC: https://portal.mrcmekong.org/monitoring/river-monitoring-telemetry). Other data, including water discharge and water levels in the mainstream of the VMD, are recorded hourly, with daily maximum and minimum water levels collected from the Ca Mau gauge within the study area. These measurements are obtained from gauges operated by the Southern Regional Hydro-Meteorological Center (SRHMC) with water levels referenced to the Hon Dau Mean Sea Level (MSL) datum and provided by SIWRR (http://www.siwrr.org.vn/)".*

- 185-186: What are the roughness values used? Were the initial roughness values determined based on LULC data?

We appreciate the reviewer's comment, we have added the following information to line 182:

The initial roughness values were based on Manh et al., (2014), who categorized the model domain into distinct zones, assigning specific Manning's n values to each zone to represent their unique hydraulic characteristics. These values were systematically refined to achieve the best agreement between the model's predicted water levels and discharges and the observed data. Specifically, the calibration process utilized monitoring stations located along the main Mekong channel in Vietnam (Fig. 2b, Table 1). Model outputs were compared with observed water levels and discharges, and any discrepancies prompted adjustments to the zone-specific roughness values. The model was iteratively rerun until optimal results were achieved. The final calibrated Manning's n values are detailed in Table 3.

**Table 3.** The Manning roughness coefficient (n) is categorized based on different zones using in the modelling. The Manning roughness in the channel changing for different cross-sections within the same zone

| Zone | Description | Manning's coefficient (n) |
| --- | --- | --- |
| 1 | Mekong River: Kratie to Phnom Penh | 0.032 |
| 2 | Mekong River: Phnom Penh to Tan Chau | 0.032 to 0.027 |
| 3 | Mekong River: Tan Chau to My Thuan | 0.027 to 0.025 |
| 4 | Mekong River: My Thuan to River months | 0.025 to 0.016 |
| 5 | Bassac River: Phnom Penh to Chau Doc | 0.032 |
| 6 | Bassac River: Chau Doc to Can Tho | 0.032 to 0.025 |
| 7 | Bassac River: Can Tho to River months | 0.025 to 0.017 |
| 8 | Side channels | 0.033 |
| 9 | Floodplain | 0.033 |

**Results**

- 340: Use singular ("scenario") and not plural.

We have made the edit

- 312-429: Could the observation that extraction-induced land subsidence and SLR result in the max. increase of inundation extent and depth also relate to the fact that the underlying data considered provide projections for the longest time scales? This potential should be discussed in the discussion section.

We thank the reviewer for their comment. We have included this in the discussion section, which now reads:

Finally, the inundation scenarios in this study are based on long-term projections that consider the cumulative impacts of land subsidence and sea level rise up to 2100. While these projections are essential for evaluating future risks and guiding long-term planning, they may not fully account for short-term fluctuations or temporary environmental changes, potentially leading to an overestimation

*of short-term impacts. Although the focus on long-term trends offers valuable insights into future scenarios, further research that includes short-term variability is needed to provide a more comprehensive understanding of the region's vulnerability to flooding and land subsidence in the near term.*

- 402-405: Why are the 4a and 4b scenarios missing in figure 5? Please add them.

We thank the reviewer for their comment. The S4_a and S4_b scenarios, which represent land subsidence, have been added to Figure 5.

- 406-408: The artefacts of the underlying DEM dataset become clearly visible in figure 6. This should be discussed in the discussion section.

We sincerely appreciate the reviewer's insightful comment. The resolution of the DEM dataset used in this study is 5x5 m, which inherently highlights variations in elevation across cells. These differences are directly reflected in the mapped inundation levels, categorized by thresholds such as <0.1 m or from 0.1 to 0.4 m. Consequently, variations in inundation area become distinctly apparent on the DEM.

However, we have addressed this point in the discussion section, where it is now thoroughly elaborated:

*The use of a 5x5 m resolution DEM in this study has provided a detailed representation of the topographical variations within the study area. However, this high level of detail also introduces certain artefacts that become apparent in Figure 6 and 7. These artefacts stem from the granularity of the DEM dataset, which captures subtle variations in elevation between adjacent cells. Such elevation differences are particularly pronounced in the mapping of inundation levels, where thresholds such as <0.1 m (unflooded; white colour, Fig. 6) and 0.1–0.4 m (flooded; green colour, Fig. 6) highlight the varying extents of inundated areas. While this granularity allows for a more precise delineation of inundation zones, it also introduces abrupt transitions in the inundation extent that may not accurately reflect real-world conditions. These artefacts are especially noticeable in flat or low-lying areas, where small discrepancies in elevation can disproportionately affect the modelled inundation extent. Despite these limitations, the 5x5 m DEM provides valuable insights into the spatial distribution of inundation areas at a resolution suitable for regional-scale assessments. The artefacts observed in this study highlight the importance of balancing DEM resolution with the study's scale and objectives to ensure accurate and reliable outcomes.*

- 419-420: The artefacts of the underlying DEM dataset become visible. Seems like as if there are also interpolation (?) artefacts in the scenarios because they look quite strange (esp. S5_b, S4_a). Please check and at least include a paragraph in the discussion section.

We appreciate the reviewer's insightful comment. As mentioned in our response to the previous point, these artefacts are a result of the high resolution of the elevation DEM used. This issue has been acknowledged and addressed in the discussion section.

**Discussion**

- 430-472: The discussion section is too short and in its current version is missing several aspects that require critical reflection. Please insert a subsection where the quality of underlying datasets as well as the limitations are discussed. This relates mainly to the DEM and land subsidence data used but may also go beyond.

We appreciate the reviewer's comment, we have revised the discussion section and expanded the Limitations to address the reviewer's concerns. Please refer to the response under point 9 for further details.

- 450-452: When adaptation and mitigation strategies are discussed, not only tidal barriers should be mentioned. What about other solutions such as sedimentation strategies that tackle the cause (i.e. elevation loss due to land subsidence)?

We appreciate the reviewer's comment. In response to point 9, we have revised the discussion section to include additional content. The section now reads:

*To mitigate the impacts of flooding caused by SLR and land subsidence in the Ca Mau Peninsula, it is crucial to implement a combination of sustainable strategies that address both the natural and anthropogenic drivers of vulnerability. A primary focus must be on reducing groundwater extraction, the leading cause of land subsidence in the region. This is critical not only for mitigating land subsidence but also for enhancing the area's resilience to the escalating impacts of climate change (Minderhoud et al., 2020). Additionally, increasing sediment flow from upstream rivers is essential for both replenishing eroded coastal areas and counteracting the effects of land subsidence. By improving sediment management in river systems, such as restoring natural sediment transport pathways and removing barriers like dams, the flow of sediment can be enhanced. Moreover, implementing water-saving technologies, such as efficient irrigation systems and the upgrading of water channels from the Bassac River, can significantly reduce the demand for groundwater in agriculture and domestic use. Moreover, developing alternative water sources, such as, rainwater harvesting, desalination, and surface water reservoirs, which can reduce dependency on groundwater, ensuring reliable water supplies during periods of drought or high demand.*

*In parallel, restoring and protecting the natural balance between saltwater and freshwater systems is essential. The creation or rehabilitation of tidal marshes and mangrove forests along the coastline can serve as natural barriers, filtering salinity, protecting against erosion and storm surges, and providing critical habitat for biodiversity while sustaining local livelihoods (Barbier et al., 2008;Temmerman et al., 2013; Jones et al., 2020; Sunkur et al., 2023; Dominicis et al., 2023). In addition, the integration of engineered infrastructure, including tidal barriers, levees, and dikes, can help prevent saltwater intrusion and manage flood risks, particularly in areas most vulnerable to the effects of SLR.*

*Community engagement is equally important in ensuring the long-term success of these strategies. Raising awareness about the negative impacts of excessive groundwater extraction and the benefits of sustainable water use can foster local participation in conservation efforts. Educational campaigns*

*and training programs can also empower local stakeholders to adopt water-saving practices, contributing to the sustainable management of this vital resource. Finally, investing in robust monitoring systems is essential for tracking groundwater levels, land subsidence rates, and water usage patterns. Accurate, real-time data can inform effective decision-making, enabling authorities to prioritize interventions and manage resources efficiently. By integrating these natural, infrastructural, and community-based strategies, the Ca Mau Peninsula can build resilience against flooding, safeguard its ecosystems, and ensure the sustainability of its agricultural and aquacultural industries, even in the face of rising seas and land subsidence.*

- 453-458: The problem of groundwater overexploitation is not discussed at all but needs to be addressed as it is the main driver of land subsidence in the region. Note that the datasets that were used to simulate scenarios S4a and S4b only consider extraction-induced land subsidence but not the total subsidence.

We thank the reviewer for their comment. In response to point 9 and the preceding point, we have revised the discussion section to include additional strategies. Please refer to our responses to point 9 and the point above.

- 459-466: As the implementation of infrastructure adds new load on the deltaic land surface, land subsidence will increase in that area. Please add this comment when the construction of the North-South Expressway is raised.

We thank the reviewer for their comment and have revised this section. It now reads:

*In addition, the study also reveals concerns regarding the region's infrastructure, particularly road construction. According to the national North-South Expressway's design specifications for the period 2021-2025 (No: 912/QĐ-BGTVT), the road crest is set 2.2 m above mean sea level in the Ca Mau area (Fig. 2). However, land subsidence projections of approximately 0.77 m by mid-century and 2.34 m by the end of the century could lower the road crest to 1.43 m and -0.14 m, respectively. These reductions may not ensure adequate protection, especially considering predicted maximum water levels of approximately 1.59 m by 2050 and 2.20 m by 2100. This assessment is based on Vietnam's road construction standards (TCVN 4054: 2005). Furthermore, the development of infrastructure contributes additional pressure to the deltaic land surface, exacerbating land subsidence in these areas.*

- 471-472: The implementation only of the measures mentioned in this discussion does not allow for this conclusion as they rather tackle symptoms than the causes. Measures that tackle subsidence-induced elevation loss need to be discussed.

We greatly appreciate the reviewer's comment. In response to point 9, we have updated the discussion section to incorporate additional measures. Please refer to our responses to point 9.

**Conclusions**

- 475: It is exposure, not vulnerability as this study does not include any social or socio-economic data.

We have made the necessary edits, and line 457 now read:

*The Ca Mau peninsula is identified as one of the most promising economic hubs for agriculture and aquaculture within both the Mekong Delta region and Vietnam as a whole. Nonetheless, these areas also experience considerable flood risk*

- 483-485: Consider previous comments on the doubled consideration of land subsidence in the combined scenarios as SLR projections from the NASA SLR projection tool include a VLM component.

We sincerely appreciate the reviewer's thoughtful feedback. We would like to clarify that land subsidence in this study is driven by groundwater extraction scenarios, and sea level rise (SLR) projections are treated as separate factors in our analysis. While the NASA SLR projection tool does include a Vertical Land Motion (VLM) component, we ensure that land subsidence is considered independently in our scenarios. This distinction allows us to avoid any potential double-counting of subsidence effects.

Furthermore, we would like to emphasize that the SLR future scenarios presented in our study, while inherently uncertain, are grounded in realistic projections based on the best available data. Such as, given the significant uncertainty surrounding ice-sheet dynamics and their potential response to global warming, global mean sea level (GMSL) projections could rise by as much as 2.5 m by 2100, with the possibility of exceeding 15 m by 2300 (Sweet et al., 2017; Hall et al., 2019; IPCC, 2023). These uncertainties are particularly relevant for long-term projections and highlight the need for a broad range of scenarios to reflect various possible future outcomes.

We have revised the manuscript to clearly highlight this distinction and have updated line 185 for improved transparency. It now reads:

*It is note that these projections do not solely rely on future predictions of sea-level rise, but also consider the dynamic processes involved. For instance, the SLR projections derived from the NASA SLR projection tool already account for Vertical Land Motion, including land subsidence, which could impact the regional SLR rates. To avoid potential double-counting, we have ensured that land subsidence is treated as a separate factor in our analysis, accounting for localized variations based on independent datasets. These adjustments are crucial for accurately representing the local context and ensuring that the impacts of subsidence are not inadvertently overestimated.*

*Hall, J. A., Weaver, C. P., Obeysekera, J., Crowell, M., Horton, R. M., Kopp, R. E., Marburger, J., Marcy, D. C., Parris, A., Sweet, W. V., Veatch, C., & White, K. D. (2019). Rising Sea Levels : Helping Decision-Makers Confront the Inevitable. Coastal Management, 47, 127–150. https://doi.org/10.1080/08920753.2019.1551012*
*IPCC. (2023). Climate change2023 Synthesis Report, Summary for Policymakers. Contribution of Working Groups I, II and III to the Sixth Assessment Report of the Intergovernmental Panel on Climate Change. https://doi.org/10.59327/IPCC/AR6-9789291691647.001*
*Sweet, W. V., Kopp, R. E., Weaver, C. P., Obeysekera, J., Horton, R. M., Thieler, E. R., & Zervas, C. (2017). Global and regional sea level rise scenarios for the united states. NOAA Technical Report NOS CO-OPS 083.*

- 500-505: Note that the adaptation measures listed here do not address the cause and whether the mangrove ecosystem can be sustained in the future.

We sincerely thank the reviewer for their helpful comments. In response to point 9, we have revised the discussion section to include additional strategies. Please refer to our responses to point 9.

---

## Author Response (AR3)

**Review response for manuscript entitled "Assessment of coastal inundation triggered by multiple drivers in Ca Mau Peninsula, Vietnam"**

We sincerely thank the editorial team and reviewers for their dedicated time and thorough evaluation of our manuscript. We are grateful for the additional feedback provided on this revised version and deeply appreciate the opportunity to revise and improve the manuscript based on the reviewers' insightful comments.

Below, we present a detailed, point-by-point response to the reviewers' remarks (shown in blue text), along with our replies (in black text) and descriptions of the corresponding changes made to the manuscript (in black italicized text).

**Report #1 ( Referee #2)**

I appreciate the authors' efforts in addressing reviewers' comments. However, there are still a couple of grammar issues in the abstract that need to be corrected.

1) Line 23: Please change "analyzing" to "analyze".

We thank the reviewer for pointing this out. We have corrected this grammatical error in the revised version.

2) Line 30: Please change "is" to "are".

We thank the reviewer for pointing this out. We have corrected this grammatical error in the revised version.

We deeply appreciate the reviewer's thoughtful feedback and ongoing support of our manuscript. We greatly appreciate the time and effort invested in reviewing both the manuscript and our responses, and we are pleased that the revisions have been deemed appropriate.

**Report #2 (Referee #3)**

This study makes a meaningful contribution to understanding the compounding effects of multiple flood drivers in a very localized Mekong Delta region. Authors demonstrate that the main drivers of flooding is not the Mekong River but the combination of other scenarios involving land subsidence and sea level rise. Below are some minor comments for authors to consider in terms of enhancing understanding and clarity for readers.

We sincerely thank the reviewer for their valuable feedback and recognition of the importance of our work in addressing compound flooding under future scenarios in coastal zones. We greatly appreciate the thoughtful comments, which have helped us further refine and improve the clarity and impact of the manuscript.

1. Abstract: Authors may consider explicitly noting that they are assessing the impact of "compounding" hazards in the region. In addition, the main text states that a "key contribution of the study is the updating of datasets" for re-calibration of the model. If this was a major effort that also significantly

improved the performance of the model, this is worth mentioning in the abstract (and show how performance was improved in supplementary).

We sincerely thank the reviewer for their insightful comment. In response, we have revised the abstract (lines 22–27) to explicitly highlight the assessment of *compounding* hazards and the model improvements. The revised text now reads:

"In this study, we assess the impact of compounding hazards by developing regional inundation maps and analysing flood dynamics in the CMP using a large-scale hydrodynamic model encompassing the entire VMD. The model was enhanced with updated bathymetric data for major river channels, along with synchronized information on the dyke across the VMD from the 2018–2019 period, resulting in a substantial performance improvement. It was then applied across multiple future scenarios based on both individual drivers and their combinations, representing a wide but plausible range of anthropogenic and climate changes"

2. Figure 3: the skewed tail at the two downstream monitoring stations implies that the model consistently under predicts reductions in the water level and discharge rates during the dry season at these locations. Was this expected?

We thank the reviewer for their insightful feedback. It is correct that the skewed tail observed at the two downstream monitoring stations (My Thuan on the Mekong River and Can Tho on the Bassac River) during periods of low discharge and water levels indicates the model tends to underpredict reductions in water level and discharge during the dry season (Fig.3, manuscript paper). However, our study primarily focuses on flood hazard assessment based on maximum water levels during flooding events to develop inundation maps. Therefore, discrepancies during low-flow periods have limited impact on the key results.

Furthermore, the differences between observed and simulated data during the lowest water levels are minimal, around tens of centimetres at the Can Tho station, which is near our study area (Ca Mau Peninsula), in both calibration and validation steps. (see Fig.3, Can Tho row, water level column, manuscript paper). Additionally, the Nash-Sutcliffe Efficiency (NSE) and coefficient of determination ( $R^2$ ) values demonstrate strong overall model performance throughout the dry season in both calibration and validation. In response to the reviewer's suggestion, we have revised the text in the 'Model calibration and validation' section. Please refer to our response to Point 3 below for further details

3. Figure 3 and Table 2: data for the Ca Mau station is missing: The Ca Mau station is the only gauge that falls within this study region (based on Fig 1). More details about model performance at this station seems warranted, building on the reference in the main text (line 226). Is the 12% discrepancy between simulation and observed value at this station is due to fluctuating water demand from aquaculture in the region?

We thank the reviewer for this insightful feedback. In response, we have collected additional water level data for the Ca Mau region, now using hourly measurements rather than only daily maximum and minimum values as in the previous version. The calibration and validation section has been revised to better reflect the model's performance across the entire Vietnamese Mekong Delta, with a particular focus on the Ca Mau station.

Regarding the previously reported 12% discrepancy between simulated and observed values at Ca Mau, we have removed this deviation figure due to the absence of a clear benchmark. Instead, we highlight that the difference between maximum simulated and observed water levels is on the order of centimeters for both calibration and validation periods, which suggests that the small difference is acceptable.

We also agree with the reviewer's suggestion that this discrepancy may be influenced by fluctuating water demand from aquaculture and agriculture in the region, which are not explicitly included in the model. However, the dense network of side channels in the study area (see Figure 1) likely limits the overall impact of these demands on water levels.

**The revised line 210- 227 now reads:**

"Minor overestimations are observed during low-flow periods at My Thuan (Mekong River) and Can Tho (Bassac River), with deviations of approximately 20 cm and 10 cm, respectively (Fig. 3a). However, these discrepancies are relatively small when compared to the daily water level fluctuation of about 2.5 m at these stations during dry season. Given that the primary objective of this study is to generate maximum inundation maps for flood hazard assessment, slight inaccuracies during low-water conditions have minimal influence on the overall outcomes.

Across the study area, the model also demonstrates strong performance at the Ca Mau gauging station. The simulated maximum water level is approximately 0.92 m, slightly overestimating the observed value of 0.84 m by around 0.08 m. This small discrepancy may be attributed to the exclusion of local water extraction for agriculture and aquaculture in the model. However, given the region's dense network of side channels, the influence of such withdrawals on overall water levels is likely minimal. This small difference is negligible compared to the daily water level fluctuation of around 2.5 m in the area. Importantly, both the NSE and R2 values demonstrate excellent agreement between the simulated and observed water level data, further confirming the model's capability to accurately represent the dynamics of water levels in the system, making it a valuable tool for flood hazard prediction and management, particularity for the study area.

For the validation stage, the results reveal a consistently strong agreement between the simulated outcomes and the corresponding observed data across the VMD gauging stations (Fig. 3b and Table 2). There is a persistent pattern of high agreement in water level values and an overall strong agreement in terms of water discharge. At the Ca Mau gauging station, the simulated maximum water level was 0.86 m, compared to the observed value of 0.77 m, an overestimation of only about 0.09 m indicating good model performance."

**4. Does it make sense that the Manning roughness coefficients are calibrated for a wet year (2018) and validated on a dry year (2016)?**

We thank the reviewer for this feedback. To the best of our knowledge, this approach is appropriate. The calibration aims to evaluate the robustness of model parameters, specifically, the hydraulic roughness coefficients (Manning's n), under certain hydrological conditions, followed by validation to test the model's performance under different hydrological scenarios. We calibrated the model using data from a high-flow year (2018), which aligns with the primary objective of this study is to assess flood hazards. The resulting optimal roughness values were then validated using data from a low-water flux year (2016) to evaluate the model's performance under low-flow conditions. This strategy ensures the model's reliability across a broad range of hydraulic condition and supports its overall effectiveness.

5. Deviation values: It would be worth showing the deviation values in a Table. Can authors provide a sense of how good the deviation values of 10-12% are based on literature? Authors state that this is a 'satisfactory level of agreement'; readers would benefit from knowing a benchmark value.

We thank the reviewer for this feedback. Kindly refer to the response to Point 3 above for detailed information.

6. Sea Level Rise vs Land Subsidence: Authors note they avoid potential double-counting by ensuring land subsidence is a separate factor (line 305), but it's unclear what are the implications for interpreting this scenario: should readers understand Scenario 5 then as the combined influence of subsidence and sea level rise?

We thank the reviewer for this insightful comment. We acknowledge that there is a complex relationship between sea level rise (SLR) and land subsidence in coastal regions. Land subsidence can result from both natural processes, such as sediment compaction and tectonic activity, and anthropogenic drivers, especially groundwater extraction. (Minderhoud et al., 2017; Karlsrud et al., 2020).

However, it is important to highlight that the sea level rise projections used in this study for the coastal Vietnamese Mekong Delta—sourced from NASA's SLR Projection Tool—already account for Vertical Land Motion, meaning that they partially include the effects of land subsidence. Therefore, it is crucial to clearly distinguish the components included in each scenario to avoid any potential double-counting.

In our study, we isolate the land subsidence scenarios (lines 276–280, manuscript), which state that "Delta Subsidence (S4\_a, S4\_b): These scenarios evaluate the changes in inundation associated with land subsidence in the projected future. It is important to note that, we only assess the impact of groundwater extraction-induce land subsidence and does not include other contributing factors such as natural subsidence, tectonic movements, or other human activities (Minderhoud et al., 2017; Zoccarato et al., 2018); Karlsrud et al., 2020). This focus is due to the fact that land subsidence in the CMP is primarily driven by groundwater extraction (Minderhoud et al., 2017; Karlsrud et al., 2020)".

That mean the land subsidence scenarios used here focus only on groundwater extraction-induced subsidence, as the fact that land subsidence in the CMP is primarily driven by groundwater extraction. We explicitly exclude other sources of vertical land movement (e.g., natural subsidence, tectonics) in

these scenarios to ensure that our assessment does not overlap with what is already embedded in the SLR projections.

In contrast, the sea level rise scenarios (line 298 and onward) consider only the rise in mean sea level as projected by climate models, including natural vertical land movement components, but excluding localized human-induced subsidence. This distinction is critical to ensure the effects of groundwater extraction are assessed independently in our land subsidence scenario.

To further reduce ambiguity, we have emphasized this clarification again in the Discussion section, where we now explicitly state:

"Regarding land subsidence scenarios, this study focuses solely on the impact of groundwater extraction, excluding other factors contributing to land subsidence, such as natural subsidence, tectonic movements, or other human activities."

In response to the reviewer's comment and to prevent potential misinterpretation, we have revised the text at line 301, revised manuscript. It now reads: "To prevent potential double-counting, we have treated land subsidence as a separate factor in our analysis, considering only subsidence driven by groundwater extraction (see Delta Subsidence Scenarios: S4\_a, S4\_b). In contrast, the SLR scenarios already incorporate land subsidence associated with Vertical Land Motion. Following this, the values representing sea-level rise scenarios are incorporated into the time series of tidal levels from the year 2018, functioning as downstream boundaries for the model scenarios."

- \*Karlsrud, K., Tunbridge, L., Khanh, N. Q., & Dinh, N. Q. (2020). Preliminary results of land subsidence monitoring in the Ca Mau Province. Proceedings of the International Association of Hydrological Sciences, 382, 111–115. https://doi.org/10.5194/piahs-382-111-2020
- Minderhoud, P. S. J., Erkens, G., Pham, V. H., Bui, V. T., Erban, L., Kooi, H., & Stouthamer, E. (2017). Impacts of 25 years of groundwater extraction on subsidence in the Mekong delta, Vietnam. Environmental Research Letters, 12. https://doi.org/10.1088/1748-9326/aa7146
- Zoccarato, C., Minderhoud, P. S. J., & Teatini, P. (2018). The role of sedimentation and natural compaction in a prograding delta: insights from the mega Mekong delta, Vietnam. Scientific Reports, 8(1), 1–12. https://doi.org/10.1038/s41598-018-29734-7

**7. Figures 4-6: It would be helpful to have short scenario names as the titles for each panel or provide a legend for readers.**

We thank the reviewer for this insightful comment. In response, we have revised Figures 4–6 to include a legend that presents the scenario names in a shortened format, as suggested.